# COUNTERFACTUAL PLANS
# UNDER DISTRIBUTIONAL AMBIGUITY

**Ngoc Bui, Duy Nguyen, Viet Anh Nguyen**
VinAI Research, Vietnam

## ABSTRACT

Counterfactual explanations are attracting significant attention due to the flourishing applications of machine learning models in consequential domains. A counterfactual plan consists of multiple possibilities to modify a given instance so that the model's prediction will be altered. As the predictive model can be updated subject to the future arrival of new data, a counterfactual plan may become ineffective or infeasible with respect to the future values of the model parameters. In this work, we study the counterfactual plans under model uncertainty, in which the distribution of the model parameters is partially prescribed using only the first- and second-moment information. First, we propose an uncertainty quantification tool to compute the lower and upper bounds of the probability of validity for any given counterfactual plan. We then provide corrective methods to adjust the counterfactual plan to improve the validity measure. The numerical experiments validate our bounds and demonstrate that our correction increases the robustness of the counterfactual plans in different real-world datasets.

## 1 INTRODUCTION

Machine learning models, thanks to their superior predictive performance, are blooming with increasing applications in consequential decision-making tasks. Along with the potential to help make better decisions, current machine learning models are also raising concerns about their explainability and transparency, especially in domains where humans are at stake. These domains span from loan approvals (Siddiqi, 2012), university admission (Waters & Miikkulainen, 2014) to job hiring (Ajunwa et al., 2016). In these applications, it is instructive to understand why a particular algorithmic decision is made, and counterfactual explanations act as a useful toolkit to comprehend (black-box) machine learning models (Wachter et al., 2017). Counterfactual explanation is also known in the field of interpretable machine learning as contrastive explanation (Miller, 2018; Karimi et al., 2020b) or recourse (Ustun et al., 2019). A counterfactual explanation suggests how an instance should be modified so as to receive an alternate algorithmic outcome. As such, it could be used as a suggestion for improvement purposes. For example, a student is rejected from graduate study, and the university can provide one or multiple counterfactuals to guide the applicant for admission in the following year. A concrete example may be of the form "get a GRE score of at least 325" or "get a 6-month research experience".

In practice, providing a counterfactual plan consisting of multiple examples is highly desirable because a single counterfactual to every applicant with the same covariates may be unsatisfactory (Wachter et al., 2017). Indeed, the covariates can barely capture the intrinsic behaviors, constraints, and unrevealed preferences of the person they represent so that the users with the same features may have different preferences to modify their input. As a consequence, a pre-emptive design choice is to provide a "menu" of possible recourses, and let the applicant choose the recourse that fits them best. Viewed in this way, a counterfactual plan has the potential to increase satisfaction and build trust among the stakeholders of any machine learning application.

Constructing a counterfactual plan, however, is not a straightforward task because of the many competing criteria in the design process. By definition, the plan should be valid: by committing to *any* counterfactual in the plan, the application should be able to flip his current unfavorable outcome to a favorable one. However, each possibility in the plan should be in the proximity of the covariates

of the applicant so that the modification is actionable. Further, the plan should consist of a diverse range of recourses to accommodate the different tastes and preferences of the population.

Russell (2019) propose a mixed-integer programming method to generate a counterfactual plan for a linear classifier, in which the diversity is imposed using a rule-based approach. In Dandl et al. (2020), the authors propose a model-agnostic approach using a multi-objective evolutionary algorithm to construct a diverse counterfactual plan. More recently, Mothilal et al. (2020) use the determinantal point process to measure the diversity of a plan. The authors then formulate an optimization problem to find the counterfactual plan that minimizes the weighted sum of three terms representing validity, proximity, and diversity.

A critical drawback of the existing works is the assumption of an invariant predictive model, which often fails to hold in practical settings. In fact, during a turbulent pandemic time, it is difficult to assume that the demographic population of students applying for postgraduate studies remain unchanged. And even in the case that the demography remains unchanged, special pandemic conditions such as hybrid learning mode or travel bans may affect the applicants' package, which in turn leads to fluctuations of the covariate distribution in the applicant pool.

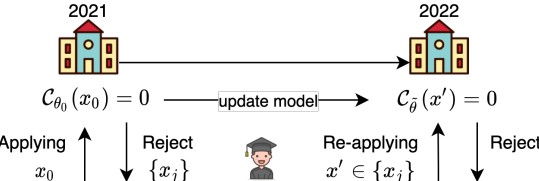

Figure 1: A student applies in Year 2021 and receives an unfavorable admission outcome. The student implements one of the recommended recourse $x'$ chosen from the counterfactual plan $\{x_j\}$ and re-applies in Year 2022. However, the outcome is again unfavorable because of the change in the model parameters $\tilde{\theta}$.

These shifts in the data are channeled to the shift in the parameters of the predictive model: when the machine learning models are re-trained or re-calibrated with new data, their parameters also change accordingly (Venkatasubramanian & Alfano, 2020). This raises an emerging concern because the counterfactual plan is usually designed to be valid to only the *current* model, but that is not enough to guarantee any validity on the *future* models. Thus, the counterfactual plan carries a promise of a favorable future outcome, nevertheless, this promise is fragile.

It is hence reasonable to demand the counterfactual plan to be robust with respect to the shift of the parameters. Pawelczyk et al. (2020) study the sparsity of counterfactuals and its non-robustness under different fixed models (predictive multiplicity). Rawal et al. (2020) consider the counterfactual plan problem and describe several types of model shift related to the correction, temporal, and geospatial shift from data. They also study the trade-off between the recourse proximity and its validity regarding the model updates. Most recently, Upadhyay et al. (2021) leverage robust optimization to generate a counterfactual that is robust to some constrained perturbations of the model's parameters. However, both works consider only the single counterfactual settings.

**Contributions.** We study the many facets of the counterfactual plans with respect to random future model parameters. We focus on a linear classification setting and we prescribe the random model parameters only through the first- and second-moment information. We contribute concretely

1. a diagnostic tool to assess the validity of a counterfactual plan. It provides a lower and upper bound on the probability of joint validity of a given plan subject to uncertain model parameters.

2. a correction tool to improve the validity of a counterfactual plan, while keeping the modifications to each counterfactual at a minimal level. The corrections are intuitive and admit closed-form expression.

3. a COunterfactual Plan under Ambiguity (COPA) framework to construct a counterfactual plan which explicitly takes the model uncertainty into consideration. It minimizes the weighted sum of validity, proximity, and diversity terms, and can be solved efficiently using gradient descents.

Each of our above contributions is exposed in Section 2, 3 and 4, respectively. In Section 5, we conduct experiments on both synthetic and real-world datasets to demonstrate the efficiency of our corrections and of our COPA framework. All proofs can be found in the appendix.

**General setup.** Consider a covariate space $\mathbb{R}^d$ and a linear binary classification setting. Each linear classifier can be parametrized by $\theta \in \mathbb{R}^d$ with decision output $\mathcal{C}_\theta(x) = 1$ if $\theta^\top x \geq 0$, and

0 otherwise, where 0 represents an unfavorable outcome. Note that we omit the bias term to avoid clutter, taking the bias term into account can be achieved by extending the dimension of $x$ and $\theta$ by an extra dimension. A counterfactual plan is a set of $J$ counterfactual explanations $\{x_j\}_{j=1,\dots,J}$, and we denote $\{x_j\}$ for short. When $J = 1$, we have a single counterfactual explanation problem, which is the subject of recent works (Ustun et al., 2019; Karimi et al., 2020a; Upadhyay et al., 2021).

Next, we define the joint validity of a counterfactual plan.

**Definition 1.1** (Joint validity). *A counterfactual plan $\{x_j\}$ is valid with respect to a realization $\theta$ if $\mathcal{C}_\theta(x_j) = 1$ for all $j = 1, \dots, J$.*

**Notations.** We use $\mathbb{S}^d_{++}$ ($\mathbb{S}^d_+$) to denote the space of symmetric positive (semi)definite matrices. For any $A \in \mathbb{R}^{m \times m}$, the trace operator is $\mathrm{Tr}\left[A\right] = \sum_{i=1}^d A_{ii}$. For any integer $J$, $[J] \triangleq \{1, \dots, J\}$.

## 2 VALIDITY BOUNDS OF COUNTERFACTUAL PLANS

In this section, we propose a diagnostic tool to benchmark the validity of a pre-computed counterfactual plan $\{x_j\}$. We model the random model parameters $\tilde{\theta}$ with a nominal distribution $\widehat{\mathbb{P}}$. Instead of making a strong assumption on a specific parametric form of $\widehat{\mathbb{P}}$ such as Gaussian distribution, we only assume that $\widehat{\mathbb{P}}$ is known only up to the second moment. More specifically, we assume that under $\widehat{\mathbb{P}}$, $\tilde{\theta}$ has a nominal mean vector $\widehat{\mu}$ and nominal covariance matrix $\widehat{\Sigma} \in \mathbb{S}^d_{++}$.

**Definition 2.1** (Gelbrich distance). *The Gelbrich distance between two pairs $(\mu_1, \Sigma_1) \in \mathbb{R}^d \times \mathbb{S}^d_+$ and $(\mu_2, \Sigma_2) \in \mathbb{R}^d \times \mathbb{S}^d_+$ is defined as*

$$\mathbb{G}\big((\mu_1, \Sigma_1), (\mu_2, \Sigma_2)\big) \triangleq \sqrt{\|\mu_1 - \mu_2\|_2^2 + \mathrm{Tr}\left[\Sigma_1 + \Sigma_2 - 2\big(\Sigma_2^{\frac{1}{2}} \Sigma_1 \Sigma_2^{\frac{1}{2}}\big)^{\frac{1}{2}}\right]}.$$

The Gelbrich distance is closely related to the optimal transport distance between Gaussian distributions. Indeed, $\mathbb{G}\big((\mu_1, \Sigma_1), (\mu_2, \Sigma_2)\big)$ is equal to the type-2 Wasserstein distance between two Gaussian distributions $\mathcal{N}(\mu_1, \Sigma_1)$ and $\mathcal{N}(\mu_2, \Sigma_2)$ (Gelbrich, 1990). It is thus trivial that $\mathbb{G}$ is a distance on $\mathbb{R}^d \times \mathbb{S}^d_+$, and as a consequence, it is symmetric and $\mathbb{G}\big((\mu_1, \Sigma_1), (\mu_2, \Sigma_2)\big) = 0$ if and only if $(\mu_1, \Sigma_1) = (\mu_2, \Sigma_2)$. Using the Gelbrich distance to design the moment ambiguity set for distributionally robust optimization leads to many desirable properties such as computational tractability and performance guarantees (Kuhn et al., 2019; Nguyen et al., 2021a). Motivated by this idea, we first construct the following uncertainty set

$$\mathcal{U} \triangleq \{(\mu, \Sigma) \in \mathbb{R}^d \times \mathbb{S}^d_+ : \mathbb{G}((\mu, \Sigma), (\widehat{\mu}, \widehat{\Sigma})) \leq \rho\},$$

which is formally a $\rho$-neighborhood in the mean vector-covariance matrix space around the nominal moment $(\widehat{\mu}, \widehat{\Sigma})$. The ambiguity set for the distributions of $\tilde{\theta}$ is obtained by lifting $\mathcal{U}$ to generate a family of probability measures that satisfy the moment conditions

$$\mathbb{B} \triangleq \{\mathbb{Q} \in \mathcal{P} : \exists (\mu, \Sigma) \in \mathcal{U} \text{ such that } \mathbb{Q} \sim (\mu, \Sigma)\},$$

where $\mathcal{P}$ is a set of all probability measures supported on $\mathbb{R}^d$ and $\mathbb{Q} \sim (\mu, \Sigma)$ indicates that $\mathbb{Q}$ has mean vector $\mu$ and covariance matrix $\Sigma$.

The central question of this section is: If the distribution of $\tilde{\theta}$ belongs to $\mathbb{B}$, what is the probability that a given plan $\{x_j\}$ is valid? To answer this question, we define the event set $\Theta(\{x_j\})$ that contains all model parameter values that renders $\{x_j\}$ jointly valid. Under the definition of a linear model, $\Theta(\{x_j\})$ is an intersection of $J$ open hyperplanes of the form

$$\Theta(\{x_j\}) \triangleq \big\{\theta \in \mathbb{R}^d : x_j^\top \theta \geq 0 \ \forall j \in [J]\big\}. \tag{1}$$

We name $\Theta(\{x_j\})$ the set of favorable parameters. The probability of validity for a plan under a measure $\mathbb{Q}$ is $\mathbb{Q}(\tilde{\theta} \in \Theta(\{x_j\}))$. We are interested in evaluating the lower and the upper bound probability that the plan $\{x_j\}$ is valid uniformly over all distributions $\mathbb{Q} \in \mathbb{B}$. This is equivalent to quantifying the following quantities

$$\inf_{\mathbb{Q} \in \mathbb{B}} \mathbb{Q}(\tilde{\theta} \in \Theta(\{x_j\})) \quad \text{and} \quad \sup_{\mathbb{Q} \in \mathbb{B}} \mathbb{Q}(\tilde{\theta} \in \Theta(\{x_j\})).$$

In the remainder of this section, we discuss how to evaluate the bounds for these terms.

**Lower bound.** We denote by $\Theta^\circ$ the interior of the set $\Theta$, that is, $\Theta^\circ(\{x_j\}) \triangleq \left\{\theta \in \mathbb{R}^d : x_j^\top \theta > 0 \ \forall j\right\}$. Note that all the inequalities defining $\Theta^\circ(\{x_j\})$ are strict inequalities. By definition, we have $\Theta^\circ(\{x_j\}) \subset \Theta(\{x_j\})$, and hence $\inf_{\mathbb{Q}\in\mathbb{B}} \mathbb{Q}(\tilde{\theta} \in \Theta^\circ(\{x_j\})) \le \inf_{\mathbb{Q}\in\mathbb{B}} \mathbb{Q}(\tilde{\theta} \in \Theta(\{x_j\}))$. Because $\Theta^\circ(\{x_j\})$ is an open set, we can leverage the generalized Chebyshev lower bound to evaluate the minimum quantity of $\mathbb{Q}(\tilde{\theta} \in \Theta^\circ(\{x_j\}))$ over all distributions with a given mean and covariance matrix (Vandenberghe et al., 2007). Adding moment uncertainty via the set $\mathcal{U}$ is obtained by rejoining two minimization layers. The next theorem presents this result.

**Theorem 2.2** (Lower bound). *For any $\rho \in \mathbb{R}_+$, $\widehat{\mu} \in \mathbb{R}^d$ and $\widehat{\Sigma} \in \mathbb{S}_+^d$, let $L^\star$ be the optimal value of the following semidefinite program*

$$
L^\star = \begin{cases}
\inf & 1 - \sum_{j\in[J]} \lambda_j \\
\text{s.t.} & \mu \in \mathbb{R}^d, \ \Sigma \in \mathbb{S}_+^d, \ C \in \mathbb{R}^{d\times d}, \ M \in \mathbb{S}_+^d \\
& \lambda_j \in \mathbb{R}, \ z_j \in \mathbb{R}^d, \ Z_j \in \mathbb{S}^d \ \forall j \in [J] \\
& -x_j^\top z_j \ge 0, \ \begin{bmatrix} Z_j & z_j \\ z_j^\top & \lambda_j \end{bmatrix} \succeq 0 \qquad \forall j \in [J] \\
& \sum_{j\in[J]} \begin{bmatrix} Z_j & z_j \\ z_j^\top & \lambda_j \end{bmatrix} \preceq \begin{bmatrix} M & \mu \\ \mu^\top & 1 \end{bmatrix}, \ \begin{bmatrix} \Sigma & C \\ C^\top & \widehat{\Sigma} \end{bmatrix} \succeq 0, \ \begin{bmatrix} M-\Sigma & \mu \\ \mu^\top & 1 \end{bmatrix} \succeq 0 \\
& \|\widehat{\mu}\|^2 - 2\widehat{\mu}^\top \mu + \mathrm{Tr}\left[M + \widehat{\Sigma} - 2C\right] \le \rho^2.
\end{cases}
\tag{2}
$$

*Then we have $L^\star = \inf_{\mathbb{Q}\in\mathbb{B}} \mathbb{Q}(\tilde{\theta} \in \Theta^\circ(\{x_j\})) \le \inf_{\mathbb{Q}\in\mathbb{B}} \mathbb{Q}(\tilde{\theta} \in \Theta(\{x_j\}))$.*

**Upper bound.** Because $\Theta(\{x_j\})$ is a closed set, we can leverage a duality result to evaluate the maximum quantity of $\mathbb{Q}(\tilde{\theta} \in \Theta(\{x_j\}))$ over all distributions with a given mean and covariance matrix (Isii, 1960). Adding moment uncertainty via the set $\mathcal{U}$ is obtained by invoking the support function of the moment set. This result is presented in the next theorem

**Theorem 2.3** (Upper bound). *For any $\rho \in \mathbb{R}_+$, $\widehat{\mu} \in \mathbb{R}^d$ and $\widehat{\Sigma} \in \mathbb{S}_+^d$, let $U^\star$ be the optimal value of the following semidefinite program*

$$
U^\star = \begin{cases}
\inf & z_0 + \gamma(\rho^2 - \|\widehat{\mu}\|_2^2 - \mathrm{Tr}\left[\widehat{\Sigma}\right]) + q + \mathrm{Tr}\left[Q\right] \\[4pt]
\text{s.t.} & \gamma \in \mathbb{R}_+, \ z_0 \in \mathbb{R}, \ z \in \mathbb{R}^d, \ Z \in \mathbb{S}_+^d, \ q \in \mathbb{R}_+, \ Q \in \mathbb{S}_+^d, \ \lambda \in \mathbb{R}_+^J \\[4pt]
& \begin{bmatrix} \gamma I - Z & \gamma \widehat{\Sigma}^{\frac{1}{2}} \\ \gamma \widehat{\Sigma}^{\frac{1}{2}} & Q \end{bmatrix} \succeq 0, \ \begin{bmatrix} \gamma I - Z & \gamma \widehat{\mu} + z \\ \gamma\widehat{\mu}^\top + z^\top & q \end{bmatrix} \succeq 0 \\[8pt]
& \begin{bmatrix} Z & z \\ z^\top & z_0 \end{bmatrix} \succeq 0, \ \begin{bmatrix} Z & z \\ z^\top & z_0 - 1 \end{bmatrix} \succeq \sum_{j\in[J]} \lambda_j \begin{bmatrix} 0 & \frac{1}{2}x_j \\ \frac{1}{2}x_j^\top & 0 \end{bmatrix}.
\end{cases}
$$

*Then we have $\sup_{\mathbb{Q}\in\mathbb{B}} \mathbb{Q}(\tilde{\theta} \in \Theta(\{x_j\})) \le U^\star$.*

Thanks to the choice of the Gelbrich distance $\mathbb{G}$, both optimization problems in Theorems 2.2 and 2.3 are *linear* semidefinite programs, and they can be solved efficiently by standard, off-the-shelf solvers such as MOSEK to high dimensions (MOSEK ApS, 2019). Other choices of distance (divergence) are also available: for example, one may opt for the Kullback-Leibler (KL) type divergence between Gaussian distribution to prescribe $\mathcal{U}$ as in Nguyen et al. (2020) and Taskesen et al. (2021). Unfortunately, the KL type divergence entails a log-determinant term, and the resulting optimization problems are no longer linear programs and are no longer solvable using MOSEK. Equipped with $L^\star$ and $U^\star$, we have the bounds

$$L^\star \le \mathbb{Q}(\{x_j\} \text{ is a valid plan}) \le U^\star \qquad \forall \mathbb{Q} \in \mathbb{B}$$

on the validity of the counterfactual plans $\{x_j\}$ under the distributional ambiguity set $\mathbb{B}$.

**Complementary information.** The previous results show that we can compute the lower bound $L^\star$ and upper bound $U^\star$ for the probability of validity by solving semidefinite programs. We now show that the two quantities $L^\star$ and $U^\star$ are complementary to each other in a specific sense.

**Proposition 2.4** (Complementary information). *For any instance, either $L^\star = 0$ or $U^\star = 1$. More specifically, we have: (i) If $\widehat{\mu} \in \Theta(\{x_j\})$, then $U^\star = 1$, and (ii) If $\widehat{\mu} \notin \Theta(\{x_j\})$, then $L^\star = 0$.*

Because $L^\star$ and $U^\star$ are bounds for a probability quantity, they are only informative when they are different from 0 and 1. Proposition 2.4 asserts that the upper bound $U^\star$ is trivial when $\widehat{\mu} \in \Theta(\{x_j\})$, while the lower bound $L^\star$ becomes trival if $\widehat{\mu} \notin \Theta(\{x_j\})$. Next, we leverage these insights to improve the validity of a given counterfactual plan.

## 3 COUNTERFACTUAL PLAN CORRECTIONS

Given a counterfactual plan $\{x_j\}$, it may happen that $\{x_j\}$ have low probability of being valid under random realizations of the future model parameter $\tilde{\theta}$. The diagnostic tools proposed in Section 2 indicate that $\{x_j\}$ has low validity when the bounds are low, and we are here interested in correcting this plan such that the lower bounds $L^\star$ are increased. Indeed, increasing $L^\star$ guarantees higher confidence that the plan is valid, should the distribution of $\tilde{\theta}$ belongs to the ambiguity set. At this point, one may be tempted to optimize $L^\star$ directly with $\{x_j\}$ by first converting problem (2) into a maximization problem, and then jointly maximizing with $\{x_j\}$ being decision variables. Unfortunately, this approach entails bilinear terms $x_j^\top z_j$ in the constraints, and this approach is notoriously challenging to solve. We thus resort to heuristics for correction. Towards this end, the results from Proposition 2.4 suggest that there are two correction operations that we need to perform to improve the validity of the counterfactual plan:

(i) When $\widehat{\mu} \notin \Theta(\{x_j\})$, then Proposition 2.4 suggests that we should modify the plan $\{x_j\}$ so that the resulting set of favorable parameters contains $\widehat{\mu}$. We term this type of correction as a Requirement correction, and we consider one specific Requirement correction in Section 3.1.

(ii) When $\widehat{\mu} \in \Theta(\{x_j\})$, we can also modify $\{x_j\}$ to as to increase the lower bound $L^\star$. This type of correction is termed an Improvement correction because its goal is to increase the validity of the counterfactual plans. We consider the Mahalanobis Improvement correction in Section 3.2.

We emphasize that the corrections of the plan $\{x_j\}$ are designed such that the modifications to each counterfactual $x_j$ should be minimal. This is achieved by two main criteria: the correction should modify as few counterfactuals as possible, and the modification to each counterfactual should also be as small as possible.

### 3.1 REQUIREMENT CORRECTION

We propose a Requirement correction with the goal of obtaining a corrected plan $\{x'_j\}$ from the given plan $\{x_j\}$ such that $\widehat{\mu}$ lies inside (or strictly inside) the set $\Theta(\{x'_j\})$. A simple Requirement correction is to construct the corrected plan $\{x'_j\}$ by

$$\forall j \in [J]: \qquad x'_j = \begin{cases} x_j & \text{if } \widehat{\mu}^\top x_j \geq \epsilon, \\ \arg\min\{\|x - x_j\|_2 \ : \ \widehat{\mu}^\top x \geq \epsilon\} & \text{if } \widehat{\mu}^\top x_j < \epsilon, \end{cases}$$

for some $\epsilon \geq 0$. Using this rule, $x'_j$ is the smallest modification of $x_j$ measured in the Euclidean distance such that $x'_j$ is valid with $\epsilon$-margin with respect to the expected future parameter $\widehat{\mu}$. The margin $\epsilon$ adds a layer of robustness: if $\epsilon > 0$ then $\widehat{\mu}$ lies in the interior of the set $\Theta(\{x'_j\})$, while if $\epsilon = 0$ then $\widehat{\mu}$ lies on the boundary of the set $\Theta(\{x'_j\})$). Moreover, it is easy to see that in the case $\widehat{\mu}^\top x_j < \epsilon$, the resulting $x'_j$ is the Euclidean projection of $x_j$ onto the hyperplane $\widehat{\mu}^\top x_j = \epsilon$. The proposed Requirement correction admits thus the analytical form:

$$\forall j \in [J]: \qquad x'_j = x_j - \frac{\min\{0, \widehat{\mu}^\top x_j - \epsilon\}}{\|\widehat{\mu}\|_2^2} \widehat{\mu}.$$

### 3.2 MAHALANOBIS IMPROVEMENT CORRECTION

Given a plan $\{x_j\}$ such that $\widehat{\mu} \in \Theta(\{x_j\})$ and an integer $K$ between 1 and $J$, the Mahalanobis Improvement correction aims to modify $K$ out of $J$ plans to obtain the corrected plan $\{x'_j\}$. The goal of this correction is to increase the lower bound value $L^\star$ associated with the plan $\{x'_j\}$, while at the same time keeping the amount of modification as small as possible. To attain this goal, we first describe the geometric intuition behind the lower bound $L^\star$ in (2), and then leverage this intuition to generate the correction.

**Geometric intuition.** We first analyze the distribution of the random vector $\tilde{\theta}$ that attains the validity lower bound $L^\star$. To simplify the exposition, we assume that $\lambda^\star > 0$ and define $\lambda_0^\star = 1 - \sum_j \lambda_j^\star$. Following the same argument as in Vandenberghe et al. (2007, §2.2), the distribution of $\tilde{\theta}$ can be constructed as a mixture of $J + 1$ random vectors $\tilde{\theta}_j$ satisfying:

$$\forall j = 0, \ldots, J : \quad \tilde{\theta} = \tilde{\theta}_j \text{ with probability } \lambda_j^\star,$$

where for each $j = 1, \ldots, J$, we have $\mathbb{E}[\tilde{\theta}_j] = z_j^\star / \lambda_j^\star$ and $\tilde{\theta}_0$ follows a properly chosen distribution. By the validity of $z^\star$, we can verify that the location $z_j^\star / \lambda_j^\star$ lies on the hyperplane $\{\theta : x_j^\top \theta = 0\}$. Thus, we can think of $\lambda_j^\star$ as the marginal increase in the lower bound $L^\star$ if we slightly perturb $x_j$ so that the point $z_j^\star / \lambda_j^\star$ lies inside the set of favorable parameters. This observation underlies the Mahalanobis correction which we describe next.

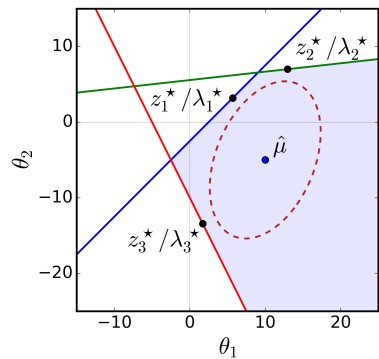

Figure 2: Illustration with $d = 2$ and $J = 3$. Shaded area is $\Theta(\{x_j\})$, dashed ellipsoid represents $(\theta - \hat{\mu})^\top \hat{\Sigma}^{-1} (\theta - \hat{\mu}) = 1$, black dots are the locations of $z_j^\star / \lambda_j^\star$.

**Correction procedure.** If we can adjust $K$ out of $J$ counterfactuals to improve the validity, then it is reasonable to modify the $K$ counterfactuals associated with the $K$ largest values of $\lambda_j^\star$, where $\lambda_j^\star$ is the optimal value of the variable $\lambda_j$ in problem (2). Without any loss of generality, assume that $\lambda_j^\star$ have decreasing values, and in this case, our correction procedure will modify the counterfactuals $x_j$ for $j = 1, \ldots, K$. Further, to correct each counterfactual, we find $x_j'$ in a $\Delta$-neighborhood of $x_j$ such that the Mahalanobis distance from $\hat{\mu}$ to the hyperplane $\{\theta : \theta^\top x_j' = 0\}$ is maximized, where the Mahalanobis distance is computed with the nominal covariance matrix $\hat{\Sigma}$. This is equivalent to solving a max-min problem

$$\begin{aligned}
x_j' = \arg\max \quad &\min_{\theta : \theta^\top x = 0} \sqrt{(\theta - \hat{\mu})^\top \hat{\Sigma}^{-1} (\theta - \hat{\mu})} \\
\text{s.t.} \quad &x \in \mathbb{R}^d, \ \|x - x_j\|_2 \leq \Delta.
\end{aligned} \tag{3}$$

The next result indicates that $x_j'$ can be found by solving a conic optimization problem.

**Theorem 3.1** (Mahalanobis Improvement correction). *The Mahalanobis correction of $x_j$ is $x_j' = v^\star / t^\star$, where $(v^\star, t^\star)$ is the optimal solution of the following conic optimization problem*

$$\min \left\{ v^\top \hat{\Sigma} v \ : \ v \in \mathbb{R}^d, \ t \in \mathbb{R}_+, \ \|v - t x_j\|_2 \leq \Delta t, \ v^\top \hat{\mu} = 1 \right\}.$$

We have specifically modified $x_j'$ in (3) with respect to the nominal mean vector and covariance matrix $(\hat{\mu}, \hat{\Sigma})$ of the random vector $\tilde{\theta}$. Alternatively, we can also use $(\mu^\star, \Sigma^\star)$, where $(\mu^\star, \Sigma^\star)$ is the optimal solution in the variable $(\mu, \Sigma)$ of (2) to form the optimization problem. Theorem 3.1 holds with the corresponding parameters $(\mu^\star, \Sigma^\star)$. Similarly, equation (4) recovers the Euclidean projection if we use an identity matrix for weighting. The conic optimization problem in Theorem 3.1 can be solved using standard off-the-shelf solvers such as Mosek (MOSEK ApS, 2019).

## 4 Counterfactual Plan Construction under Ambiguity

We propose in this section the COunterfactual Plan under Ambiguity (COPA) framework to devise a counterfactual plan that has high validity under random future model parameters. Given an input instance $x_0$, COPA builds a plan $\{x_j\}$ of $J \geq 1$ counterfactuals that balances competing objectives including proximity, diversity, and validity. We next describe each cost component.

**Proximity.** It is reasonable to ask that each counterfactual $x_j$ should be close to the input $x_0$ so that $x_j$ is actionable. We suppose that the distance between $x_0$ and $x_j$ can be measured using a function $c : \mathbb{R}^d \times \mathbb{R}^d \to \mathbb{R}$. In general, the cost $c$ is used to capture the ease of adopting the changes for a specific variable (e.g., one could barely change their height or race). The proximity of a plan $\{x_j\}$ is

simply the average distance from $x_0$ to each counterfactual in the plan. More specifically, we have

$$\text{Proximity}(\{x_j\}, x_0) \triangleq \frac{1}{J} \sum_{j=1}^{J} c(x_j, x_0). \tag{4}$$

**Diversity.** We measure the diversity of a plan using the determinant point process (Kulesza, 2012) similar to the approach in Mothilal et al. (2020). The diversity is given by:

$$\text{Diversity}(\{x_j\}) \triangleq \det(K), \text{ where } K_{i,j} = (1 + c(x_i, x_j))^{-1} \ \forall 1 \leq i, j \leq J. \tag{5}$$

Then, a plan with a larger value $\text{Diversity}(\{x_j\})$ is more diverse.

**Validity.** Given the moment information $(\widehat{\mu}, \widehat{\Sigma})$, one potential approach to compute the validity of a plan $\{x_j\}$ is to compute the value $L^\star$ in (2). However, for large covariate dimension $d$ or high number of counterfactual $J$, the semidefinite program (2) becomes time-consuming to solve and is not practical. This entails us to derive the a computationally efficient proxy for the validity of $\{x_j\}$. Towards this goal, we use the volume of the maximum-volume ellipsoid with center $\widehat{\mu}$ and covariance $\widehat{\Sigma}$ that can be inscribed in $\Theta(\{x_j\})$. Following Boyd & Vandenberghe (2004, §8.4.2), an ellipsoid with center $\widehat{\mu}$, covariance matrix $\widehat{\Sigma}$ and radius $r$ can be written in the parametric form as $\mathcal{E}_{(\widehat{\mu},\widehat{\Sigma})}(r) = \{\widehat{\Sigma}^{\frac{1}{2}} u + \widehat{\mu} : \|u\|_2 \leq r\}$. The validity of the plan $\{x_j\}$ is thus defined as

$$\text{Validity}(\{x_j\}, \widehat{\mu}, \widehat{\Sigma}) \triangleq \max\{r : r \geq 0, \ \mathcal{E}_{(\widehat{\mu},\widehat{\Sigma})}(r) \subseteq \Theta(\{x_j\})\}.$$

The next result asserts that the above validity measure can be re-expressed in closed form, which justifies its computational efficiency.

**Lemma 4.1** (Validity value). *If $\widehat{\mu} \in \Theta(\{x_j\})$, then* $\text{Validity}(\{x_j\}, \widehat{\mu}, \widehat{\Sigma}) = \min_j \widehat{\mu}^\top x_j / \|\widehat{\Sigma}^{\frac{1}{2}} x_j\|_2$.

Lemma 4.1 and the analysis in Proposition 2.4 also suggest that the counterfactual plan should satisfy $\widehat{\mu} \in \Theta(\{x_j\})$ so as to improve the validity. Similar to Section 3.1, we will impose the constraints that $\widehat{\mu}^\top x_j \geq \epsilon \ \forall j$ for some margin $\epsilon \geq 0$ for validity purposes.

**COPA framework.** Our COPA framework finds the counterfactual plan that minimizes the weighted sum of the proximity, the diversity and the validity measure. More precisely, the COPA counterfactual plan is the minimizer of

$$\begin{aligned} \min_{x_1,\ldots,x_J} \quad & \text{Proximity}(\{x_j\}, x_0) - \lambda_1 \text{Validity}(\{x_j\}, \widehat{\mu}, \widehat{\Sigma}) - \lambda_2 \text{Diversity}(\{x_j\}) \\ \text{s.t.} \quad & \widehat{\mu}^\top x_j \geq \epsilon \qquad \forall j \end{aligned} \tag{6}$$

for some non-negative parameters $\lambda_1$ and $\lambda_2$. The COPA problem (6) can be solved efficiently under mild conditions using a projected (sub)gradient descent algorithm.

A projected gradient descent algorithm can be used to solve the COPA problem (6). The gradient of the objective function of (6) can be computed using auto-differentiation. We now discuss further the projection operator. Let $\mathcal{X} \triangleq \{x \in \mathbb{R}^d : \widehat{\mu}^\top x \geq \epsilon\}$, then the feasible set of the COPA problem (6) is a product space $\mathcal{X}^J$. The projection operator $\text{Proj}_{\mathcal{X}^J}$ on the product set $\mathcal{X}^J$ is decomposable into simpler projections onto individual set $\mathcal{X}$ as $\text{Proj}_{\mathcal{X}^J}(\{x'_j\}) = \{\text{Proj}_{\mathcal{X}}(x'_1), \ldots, \text{Proj}_{\mathcal{X}}(x'_J)\}$, where each individual projection is

$$\text{Proj}_{\mathcal{X}}(x'_j) = \arg\min\{\|x - x'_j\|_2 : \widehat{\mu}^\top x \geq \epsilon\} = x'_j - \min\{0, \widehat{\mu}^\top x'_j - \epsilon\}\widehat{\mu}/\|\widehat{\mu}\|_2^2.$$

Note that the second equality above follows from the analytical formula for the Euclidean projection onto a half-space, which was previously used in Section 3.1.

## 5 Numerical Experiments

In this section, we evaluate the correctness of our validity bounds and the performance of our corrections and our COPA framework on both synthetic and real-world datasets. Our baseline for comparison is the counterfactual plan constructed from the state-of-the-art DiCE framework (Mothilal et al., 2020). Throughout the experiments, we set the number of counterfactuals to $J = 5$. For DiCE, we use the default parameters recommended in the DiCE source code. The Mahalanobis correction will use the counterfactual plan obtained by the DiCE method with $K = 3$ and the perturbation limit $\Delta$ is 0.1. In our COPA framework, we use Adam optimizer to implement Projected Gradient Descent and $\ell_2$-distance to compute perturbation cost between inputs.

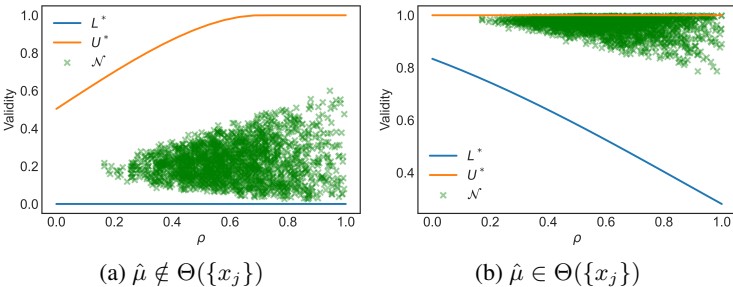

(a) $\hat{\mu} \notin \Theta(\{x_j\})$        (b) $\hat{\mu} \in \Theta(\{x_j\})$

Figure 3: The impact of the Gelbrich radius on the validity of a counterfactual plan. The vertical axis of each green point represents the empirical validity of the plan with respect to which $\tilde{\theta} \sim \mathcal{N}(\mu_g, \Sigma_g)$ and the horizontal axis is the Gelbrich distance $\mathbb{G}((\hat{\mu}, \hat{\Sigma}), (\mu_g, \Sigma_g))$.

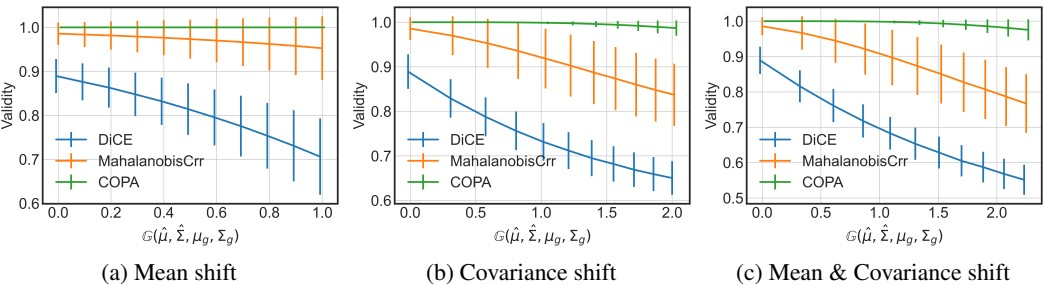

(a) Mean shift       (b) Covariance shift       (c) Mean & Covariance shift

Figure 4: The impact of shift magnitudes on the validity of the plans obtained by three algorithms.

## 5.1 SYNTHETIC DATASET

We first generate 1000 samples with two-dimensional features from two Gaussian distributions $\mathcal{N}(\mu_0, \Sigma_0)$ and $\mathcal{N}(\mu_1, \Sigma_1)$ to create a synthetic dataset. Each instance is labelled as 0 or 1 corresponding to the distribution that generated it. For the Gaussian distributions, we use similar parameters as in Upadhyay et al. (2021), where $\mu_0 = [-2, -2]^\top$, $\mu_1 = [2, 2]^\top$, $\Sigma_0 = \Sigma_1 = 0.5I$ with $I$ being the identity matrix. This dataset is then used to train a logistic classifier with the present parameter $\theta_0$. This classifier $\mathcal{C}_{\theta_0}$ is fixed for the experiments that follow.

**The impact of Gelbrich radius on the validity.** Given a counterfactual plan generated by DiCE on the classifier $\mathcal{C}_{\theta_0}$, we consider two scenarios: $\hat{\mu} \in \Theta(\{x_j\})$ and $\hat{\mu} \notin \Theta(\{x_j\})$. We choose $\hat{\mu} = \theta_0$ for the case $\hat{\mu} \in \Theta(\{x_j\})$ and $\hat{\mu} = -\theta_0$, otherwise. We also set $\hat{\Sigma} = 0.5I$. We then compute the lower and upper validity bound of this plan with respect to different Gelbrich bounds $\rho \in [0, 1]$. To evaluate the empirical validity of this plan, we simulate 1000 futures for $\tilde{\theta}$. For each future, we generate $\mu_g$ and $\Sigma_g$ randomly so that $\mathbb{G}((\hat{\mu}, \hat{\Sigma}), (\mu_g, \Sigma_g)) \leq 1$, and then we sample $10^6$ values of $\tilde{\theta} \sim \mathcal{N}_g(\mu_g, \Sigma_g)$. The empirical validity of the plan for each future is the fraction of parameter samples from the future that the prescribed plan is valid. We plot the 1000 empirical validity of the plan in Figure 3. This result is consistent with our guarantees that the validity is between the two bounds. We also observe that increasing $\rho$ loosens the validity bounds.

**The impact of degree of distribution shift on validity of a plan.** We explore the case $\hat{\mu} \in \Theta(\{x_j\})$, where $\hat{\mu} = \theta_0$ and $\hat{\Sigma} = 0.5I$, to assess the impact of distribution shift to three algorithms DiCE, MahalanobisCrr, and COPA. In this experiment, we run our COPA framework with $\lambda_1 = 2.0$, $\lambda_2 = 200.0$. To assess the performance of three algorithms, we parameterize the ground truth distribution of the future parameters $\tilde{\theta} \sim \mathcal{N}(\mu_g, \Sigma_g)$ as follows: $\mu_g = \hat{\mu} + \alpha[0, -1, 0]^\top$, $\Sigma_g = (1 + \beta)I$. Here, we simulate three types of distributional shift of the parameters $\tilde{\theta}$: (1) mean shift ($\alpha \in [0, 1], \beta = 0$), (2) covariance shift ($\alpha = 0, \beta \in [0, 3]$), and (3) mean and covariance shift ($(\alpha, \beta) \in [0, 1] \times [0, 3]$). For each shift's type, we generate 100 counterfactual plans corresponding to 100 original inputs $x_0$ and compute the empirical validity as previously described. The average and confidence range of the empirical validity are plotted in Figure 4. This result shows the tendency of decreasing validity measure of all algorithms when increasing the Gelbrich distance between estimate and ground truth distribution. However, COPA shows stability and robustness for

all shift types. The validity of DiCE deteriorates when the ground truth distribution is far from $\theta_0$. Meanwhile, MahalanobisCrr increases the robustness of the plans obtained by DiCE significantly.

## 5.2 REAL-WORLD DATASETS

In this experiment, we evaluate the robustness of the counterfactual plans obtained by three frameworks on the real datasets. We use three real-world datasets: *German Credit* (Dua & Graff, 2017; Groemping, 2019), *Small Bussiness Administration (SBA)* (Li et al., 2018), and *Student performance* (Cortez & Silva, 2008). Each dataset contains two sets of data (the present data - $D_1$ and the shifted data $D_2$). The shifted dataset $D_2$ could capture the correction shift (German credit), the temporal shift (SBA), or the geospatial shift (Student). More details for each dataset are provided in Appendix.

**Experimental settings.** For each present dataset $D_1$, we train a logistic classifier $\mathcal{C}_{\theta_0}$ with parameter $\theta_0$ on 80% of instances of the dataset and fix this classifier to construct counterfactual plans in whole experiment. We generate 100 counterfactual plans for 100 original inputs and report the average values of our evaluation metrics. To estimate $\widehat{\mu}$ and $\widehat{\Sigma}$, we train 1000 different classifiers from the present dataset $D_1$ (each is trained on a random set containing 50% instances of $D_1$), then use the empirical mean and covariance matrix of the parameter. We set Gelbrich radius $\rho = 0.01$.

**Metrics.** To compute the empirical validity in the shift dataset $D_2$, we sample 50% instances of $D_2$ 1000 times to train 1000 different logistic classifiers. We then report the empirical validity of a plan as the fraction of the classifiers with respect to which the plan is valid. We also use the lower validity bound as a metric for evaluating the robustness of a plan. We use the formula in (4) and (5) to measure the proximity and diversity of a counterfactual plan.

Table 1: Performance of competing algorithms on real world datasets. For Proximity, lower is better. For Diversity, $L^\star$ and Validity, higher is better. Bold indicate the best performance for each dataset.

| Dataset | Method | Proximity | Diversity | $L^*$ | Empirical Validity |
|---------|--------|-----------|-----------|-------|--------------------|
| Correction | DiCE | $0.986 \pm 0.324$ | $0.072 \pm 0.050$ | $0.649 \pm 0.073$ | $0.996 \pm 0.008$ |
| | MahalanobisCrr | $1.002 \pm 0.323$ | $0.064 \pm 0.047$ | $0.750 \pm 0.064$ | $0.999 \pm 0.003$ |
| | COPA ($\lambda_1 = 0.2; \lambda_2 = 2.0$) | $\mathbf{0.916} \pm 0.178$ | $0.017 \pm 0.058$ | $0.944 \pm 0.168$ | $0.997 \pm 0.018$ |
| | COPA ($\lambda_1 = 0.5; \lambda_2 = 5.0$) | $1.154 \pm 0.253$ | $0.114 \pm 0.101$ | $\mathbf{0.946} \pm 0.040$ | $\mathbf{1.000} \pm 0.000$ |
| | COPA ($\lambda_1 = 1.0; \lambda_2 = 10.0$) | $1.351 \pm 0.166$ | $\mathbf{0.225} \pm 0.045$ | $0.911 \pm 0.022$ | $\mathbf{1.000} \pm 0.000$ |
| Temporal | DiCE | $2.037 \pm 0.470$ | $0.089 \pm 0.057$ | $0.946 \pm 0.014$ | $0.801 \pm 0.061$ |
| | MahalanobisCrr | $2.014 \pm 0.473$ | $0.085 \pm 0.055$ | $0.966 \pm 0.007$ | $0.945 \pm 0.062$ |
| | COPA ($\lambda_1 = 0.2; \lambda_2 = 2.0$) | $\mathbf{1.831} \pm 0.139$ | $0.253 \pm 0.026$ | $0.994 \pm 0.000$ | $1.000 \pm 0.000$ |
| | COPA ($\lambda_1 = 0.5; \lambda_2 = 5.0$) | $1.966 \pm 0.112$ | $0.363 \pm 0.012$ | $\mathbf{0.995} \pm 0.000$ | $\mathbf{1.000} \pm 0.000$ |
| | COPA ($\lambda_1 = 1.0; \lambda_2 = 10.0$) | $2.010 \pm 0.124$ | $\mathbf{0.380} \pm 0.006$ | $0.995 \pm 0.000$ | $\mathbf{1.000} \pm 0.000$ |
| Geospatial | DiCE | $\mathbf{1.486} \pm 0.325$ | $\mathbf{0.136} \pm 0.044$ | $0.549 \pm 0.307$ | $0.408 \pm 0.363$ |
| | MahalanobisCrr | $1.497 \pm 0.325$ | $0.126 \pm 0.044$ | $0.864 \pm 0.117$ | $0.757 \pm 0.284$ |
| | COPA ($\lambda_1 = 0.2; \lambda_2 = 2.0$) | $1.779 \pm 0.352$ | $0.052 \pm 0.047$ | $\mathbf{0.998} \pm 0.000$ | $\mathbf{1.000} \pm 0.000$ |
| | COPA ($\lambda_1 = 0.5; \lambda_2 = 5.0$) | $1.882 \pm 0.353$ | $0.089 \pm 0.032$ | $0.998 \pm 0.000$ | $\mathbf{1.000} \pm 0.000$ |
| | COPA ($\lambda_1 = 1.0; \lambda_2 = 10.0$) | $1.926 \pm 0.349$ | $0.109 \pm 0.024$ | $0.997 \pm 0.000$ | $\mathbf{1.000} \pm 0.000$ |

**Results.** The results in Table 1 show that our COPA framework achieves the highest empirical validity, $L^*$, and diversity (especially when increasing $\lambda_2$) in all evaluated datasets. Comparing DiCE and Mahalanobis correction, we can observe that the trade-off of proximity and diversity of Mahalanobis correction is relatively small as compared to its improvement in terms of validity.

## 6 CONCLUSION

This paper studies the problem of generating counterfactual plans under the distributional shift of the classifier's parameters given the fact that the classification model is usually updated upon the arrival of new data. We propose an uncertainty quantification tool to compute the bounds of the probability of validity for a given counterfactual plan, subject to uncertain model parameters. Further, we introduce a correction tool to increase the validity of the given plan. We also propose a COPA framework to construct a counterfactual plan by taking the model uncertainty into consideration. The experiments demonstrate the efficiency of our methods on both synthetic and real-world datasets. Further extensions, notably to incorporate nonlinearities, are presented in the appendix.

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

## A PROOFS

### A.1 PROOFS OF SECTION 2

*Proof of Theorem 2.2.* For any $(\mu, \Sigma) \in \mathbb{R}^d \times \mathbb{S}_+^d$, let

$$\mathcal{P}(\mu, \Sigma) \triangleq \{\mathbb{Q} : \mathbb{E}_{\mathbb{Q}}[\tilde{\theta}] = \mu, \ \mathbb{E}_{\mathbb{Q}}[\tilde{\theta}\tilde{\theta}^\top] = \mu\mu^\top + \Sigma\}$$

denote the set of probability measures under which the random vector $\tilde{\theta}$ has mean $\mu$ and covariance matrix $\Sigma$. The infimum probability can be decomposed as

$$\inf_{\mathbb{Q} \in \mathbb{B}} \mathbb{Q}(\tilde{\theta} \in \Theta^\circ(\{x_j\})) = \inf_{(\mu, \Sigma) \in \mathcal{U}} \inf_{\mathbb{Q} \in \mathcal{P}(\mu, \Sigma)} \mathbb{Q}(\tilde{\theta} \in \Theta^\circ(\{x_j\}))$$

$$= \begin{cases} \inf_{(\mu, \Sigma) \in \mathcal{U}} & \inf \quad 1 - \sum_{j \in [J]} \lambda_j \\ & \text{s.t.} \quad \lambda_j \in \mathbb{R}, \ z_j \in \mathbb{R}^d, \ Z_j \in \mathbb{S}^d \ \forall j \in [J] \\ & \quad -x_j^\top z_j \geq 0, \ \begin{bmatrix} Z_j & z_j \\ z_j^\top & \lambda_j \end{bmatrix} \succeq 0 \qquad \forall j \in [J] \\ & \quad \sum_{j \in [J]} \begin{bmatrix} Z_j & z_j \\ z_j^\top & \lambda_j \end{bmatrix} \preceq \begin{bmatrix} \Sigma + \mu\mu^\top & \mu \\ \mu^\top & 1 \end{bmatrix}, \end{cases}$$

where the second equality follows from Vandenberghe et al. (2007, §2). By Malagò et al. (2018, Proposition 2), we have

$$\mathbb{G}^2((\mu, \Sigma), (\hat{\mu}, \hat{\Sigma})) = \begin{cases} \min_{C \in \mathbb{R}^{d \times d}} & \|\mu - \hat{\mu}\|^2 + \mathrm{Tr}\left[\Sigma + \hat{\Sigma} - 2C\right] \\ \text{s.t.} & \begin{bmatrix} \Sigma & C \\ C^\top & \hat{\Sigma} \end{bmatrix} \succeq 0. \end{cases}$$

$$= \begin{cases} \min_{C \in \mathbb{R}^{d \times d}} & \|\hat{\mu}\|^2 - 2\hat{\mu}^\top\mu + \mathrm{Tr}\left[\Sigma + \mu\mu^\top + \hat{\Sigma} - 2C\right] \\ \text{s.t.} & \begin{bmatrix} \Sigma & C \\ C^\top & \hat{\Sigma} \end{bmatrix} \succeq 0. \end{cases}$$

Hence, by combing two infimum operators, we have

$$\inf_{\mathbb{Q} \in \mathbb{B}} \mathbb{Q}(\tilde{\theta} \in \Theta^\circ(\{x_j\})) = \begin{cases} \inf & 1 - \sum_{j \in [J]} \lambda_j \\ \text{s.t.} & \mu \in \mathbb{R}^d, \ \Sigma \in \mathbb{S}_+^d, \ C \in \mathbb{R}^{d \times d} \\ & \lambda_j \in \mathbb{R}, \ z_j \in \mathbb{R}^d, \ Z_j \in \mathbb{S}^d \ \forall j \in [J] \\ & -x_j^\top z_j \geq 0, \ \begin{bmatrix} Z_j & z_j \\ z_j^\top & \lambda_j \end{bmatrix} \succeq 0 \qquad \forall j \in [J] \\ & \sum_{j \in [J]} \begin{bmatrix} Z_j & z_j \\ z_j^\top & \lambda_j \end{bmatrix} \preceq \begin{bmatrix} \Sigma + \mu\mu^\top & \mu \\ \mu^\top & 1 \end{bmatrix} \\ & \|\hat{\mu}\|^2 - 2\hat{\mu}^\top\mu + \mathrm{Tr}\left[\Sigma + \mu\mu^\top + \hat{\Sigma} - 2C\right] \leq \rho^2, \quad \begin{bmatrix} \Sigma & C \\ C^\top & \hat{\Sigma} \end{bmatrix} \succeq 0. \end{cases}$$

In the last step, we add an auxiliary variable $M \in \mathbb{S}_+^d$ with the constraint $M = \Sigma + \mu\mu^\top$. Note that this constraint can be replaced by $M \succeq \Sigma + \mu\mu^\top$ without affecting the optimal value of the optimization problem. Using the Schur complement, this constraint is equivalent to

$$\begin{bmatrix} M - \Sigma & \mu \\ \mu^\top & 1 \end{bmatrix} \succeq 0.$$

This completes the proof. $\square$

*Proof of Theorem 2.3.* Let $\mathbb{1}_\Theta(\theta)$ be the indicator function of the set $\Theta(\{x_j\})$, that is,

$$\mathbb{1}_\Theta(\theta) = \begin{cases} 1 & \text{if } \theta \in \Theta(\{x_j\}), \\ 0 & \text{otherwise.} \end{cases}$$

By defining the loss function $\ell(\theta) = \mathbb{1}_\Theta(\theta)$ and let $\mathcal{Z}$ be the convex feasible set defined by

$$\mathcal{Z} \triangleq \left\{ z_0 \in \mathbb{R}, z \in \mathbb{R}^d, Z \in \mathbb{S}^d : z_0 + 2z^\top\theta + \langle Z, \theta\theta^\top \rangle \geq \mathbb{1}_\Theta(\theta) \quad \forall \theta \in \mathbb{R}^d \right\}.$$

Notice that $\mathcal{Z}$ is a closed and convex set because it is an intersection of uncountably many closed and convex sets. Denote the following set $\mathcal{V}$ of mean - second moment matrices that are induced by $\mathcal{U}$ by

$$\mathcal{V} \triangleq \{(\mu, M) \in \mathbb{R}^d \times \mathbb{S}_+^d : \exists (\mu, \Sigma) \in \mathcal{U} \text{ such that } (\mu, M) = (\mu, \Sigma + \mu\mu^\top)\}.$$

The support function $\delta_\mathcal{V}^*$ of the set $\mathcal{V}$ is defined as

$$\delta_\mathcal{V}^*(z, Z) = \sup\{z^\top \mu + \mathrm{Tr}\left[ZM\right] : (\mu, M) \in \mathcal{V}\}.$$

Using these notations, we now have

$$\sup_{\mathbb{Q} \in \mathbb{B}} \mathbb{Q}(\theta \in \Theta(\{x_j\})) = \sup_{(\mu,\Sigma) \in \mathcal{U}} \sup_{\mathbb{Q} \in \mathcal{P}(\mu,\Sigma)} \mathbb{E}_\mathbb{Q}[\mathbb{1}_\Theta(\tilde{\theta})] \tag{7a}$$

$$\leq \sup_{(\mu,\Sigma) \in \mathcal{U}} \inf_{(z_0,z,Z) \in \mathcal{Z}} z_0 + 2\mu^\top z + \mathrm{Tr}\left[(\Sigma + \mu\mu^\top)Z\right] \tag{7b}$$

$$= \sup_{(\mu,M) \in \mathcal{V}} \inf_{(z_0,z,Z) \in \mathcal{Z}} z_0 + 2\mu^\top z + \mathrm{Tr}\left[MZ\right]$$

$$= \inf_{(z_0,z,Z) \in \mathcal{Z}} \sup_{(\mu,M) \in \mathcal{V}} z_0 + 2\mu^\top z + \mathrm{Tr}\left[MZ\right] \tag{7c}$$

$$= \inf_{(z_0,z,Z) \in \mathcal{Z}} z_0 + \delta_\mathcal{V}^*(2z, Z),$$

where equality (7a) is from the two layer decomposition of the ambiguity set $\mathbb{B}$, and inequality (7b) is from the Isii's duality result Isii (1960). Equality (7c) follows from the Sion's minimax theorem Sion (1958) which holds because the objective function is linear in each variable and because $\mathcal{V}$ is compact by the compactness of $\mathcal{U}$ (Nguyen et al., 2021b, Lemma A.6). We thus have

$$\sup_{\mathbb{Q} \in \mathbb{B}} \mathbb{Q}(\theta \in \Theta(\{x_j\})) = \begin{cases} \inf & z_0 + \gamma(\rho^2 - \|\widehat{\mu}\|_2^2 - \mathrm{Tr}\left[\widehat{\Sigma}\right]) + q + \mathrm{Tr}\left[Q\right] \\ \text{s.t.} & \gamma \in \mathbb{R}_+, \ z_0 \in \mathbb{R}, \ z \in \mathbb{R}^d, \ Z \in \mathbb{S}^d, \ q \in \mathbb{R}_+, \ Q \in \mathbb{S}_+^d \\ & \begin{bmatrix} \gamma I - Z & \gamma\widehat{\Sigma}^{\frac{1}{2}} \\ \gamma\widehat{\Sigma}^{\frac{1}{2}} & Q \end{bmatrix} \succeq 0, \ \begin{bmatrix} \gamma I - Z & \gamma\widehat{\mu} + z \\ \gamma\widehat{\mu}^\top + z^\top & q \end{bmatrix} \succeq 0 \\ & z_0 + 2z^\top\theta + \left\langle Z, \theta\theta^\top \right\rangle \geq \mathbb{1}_\Theta(\theta) \quad \forall \theta \in \mathbb{R}^d, \end{cases}$$

where the equality follows by substituting the support function of $\mathcal{V}$ in Kuhn et al. (2019, Lemma 2). Consider now the last constraint of the above optimization problem, it is easy to see that it is equivalent to

$$\begin{cases} z_0 + 2z^\top\theta + \left\langle Z, \theta\theta^\top \right\rangle & \geq 0 \quad \forall \theta \in \mathbb{R}^d, \\ z_0 + 2z^\top\theta + \left\langle Z, \theta\theta^\top \right\rangle & \geq 1 \quad \forall \theta \in \Theta(\{x_j\}). \end{cases}$$

The first semi-infinite constraint is equivalent to the semidefinite constraints

$$Z \succeq 0, \quad \begin{bmatrix} Z & z \\ z^\top & z_0 \end{bmatrix} \succeq 0.$$

A sufficient condition for the second semi-infinite constraint is that

$$\exists \lambda \in \mathbb{R}_+^J : \quad \begin{bmatrix} Z & z \\ z^\top & z_0 - 1 \end{bmatrix} \succeq \sum_{j \in [J]} \lambda_j \begin{bmatrix} 0 & \frac{1}{2}x_j \\ \frac{1}{2}x_j^\top & 0 \end{bmatrix},$$

which holds thanks to the S-lemma Pólik & Terlaky (2007). Adding these above constraints into the optimization problem leads to the desired upper bound. This completes the proof. □

The proof of Proposition 2.4 relies on the following result on the multivariate Chebyshev inequalities, which can be found in Marshall & Olkin (1960) and Bertsimas & Popescu (2005).

**Theorem A.1** (Multivariate Chebyshev inequality). *Let $\mathcal{S}$ be a convex set, then*

$$\sup_{\mathbb{Q} \sim (\mu,\Sigma)} \mathbb{Q}(\tilde{\theta} \in \mathcal{S}) = \frac{1}{1+\kappa}, \qquad \kappa = \inf_{\theta \in \mathcal{S}} (\theta - \mu)^\top \Sigma^{-1} (\theta - \mu).$$

We are now ready to prove Proposition 2.4.

*Proof of Proposition 2.4.* If $\widehat{\mu} \in \Theta(\{x_j\})$, then it is clear that

$$\inf_{\theta \in \Theta(\{x_j\})} (\theta - \widehat{\mu})^\top \widehat{\Sigma}^{-1} (\theta - \widehat{\mu}) = 0,$$

thus we have $U^\star \geq \sup_{\mathbb{Q} \sim (\widehat{\mu}, \widehat{\Sigma})} \mathbb{Q}(\tilde{\theta} \in \Theta(\{x_j\})) = 1$ by Theorem A.1. This leads to $U^\star = 1$.

Consider the case when $\widehat{\mu} \notin \Theta(\{x_j\})$. By the hyperplane separation theorem, there exists a vector $\bar{x} \in \mathbb{R}^d$ such that $\bar{x}^\top \theta \geq 0$ for all $\theta \in \Theta(\{x_j\})$ and $\bar{x}^\top \widehat{\mu} < 0$. Let $\mathbb{T} \triangleq \{\theta : \bar{x}^\top \theta \geq 0\}$, then it is trivial that $\Theta(\{x_j\}) \subseteq \mathbb{T}$ and that $\mathbb{T}$ is a convex set. We now have

$$\inf_{\mathbb{Q} \in \mathbb{B}} \mathbb{Q}(\tilde{\theta} \in \Theta(\{x_j\})) = 1 - \sup_{\mathbb{Q} \in \mathbb{B}} \mathbb{Q}(\tilde{\theta} \notin \Theta(\{x_j\})) \leq 1 - \sup_{\mathbb{Q} \in \mathbb{B}} \mathbb{Q}(\tilde{\theta} \notin \mathbb{T}) = 1 - 1 = 0,$$

where the penultimate equality follows from Theorem A.1. This leads to $L^\star = 0$. $\qquad\square$

## A.2    PROOFS OF SECTION 3

*Proof of Theorem 3.1.* Notice that the optimal solution in $x$ should satisfy $x^\top \widehat{\mu} > 0$. Fix any value of $x \neq 0$. Consider first the inner minimization problem of (3), and associate with the equality constraint a Lagrangian dual variable $\zeta \in \mathbb{R}$, we have

$$\begin{aligned}
\min_{\theta : \theta^\top x = 0} (\theta - \widehat{\mu})^\top \widehat{\Sigma}^{-1} (\theta - \widehat{\mu}) &= \min_{\theta \in \mathbb{R}^d} \max_{\zeta \in \mathbb{R}} (\theta - \widehat{\mu})^\top \widehat{\Sigma}^{-1} (\theta - \widehat{\mu}) + 2\zeta \theta^\top x \\
&= \max_{\zeta \in \mathbb{R}} \min_{\theta \in \mathbb{R}^d} (\theta - \widehat{\mu})^\top \widehat{\Sigma}^{-1} (\theta - \widehat{\mu}) + 2\zeta \theta^\top x \\
&= \max_{\zeta \in \mathbb{R}} \; -\zeta^2 x^\top \widehat{\Sigma} x + 2\zeta \widehat{\mu}^\top x \\
&= \frac{(x^\top \widehat{\mu})^2}{x^\top \widehat{\Sigma} x},
\end{aligned}$$

where the second equality follows from convex duality result. The third equality follows from the fact that for every value of $\zeta$, the optimal solution in the variable $\theta$ is

$$\theta^\star(\zeta) = \widehat{\mu} - \zeta \widehat{\Sigma} x.$$

Moreover, the last equality follows from the optimality condition in $\zeta$ which gives $\zeta^\star = \widehat{\mu}^\top x / (x^\top \widehat{\Sigma} x)$. Because the optimal solution in $x$ should satisfy $x^\top \widehat{\mu} > 0$, problem (3) is hence equivalent to

$$\begin{aligned}
\max \quad &\frac{x^\top \widehat{\mu}}{\sqrt{x^\top \widehat{\Sigma} x}} \\
\text{s.t.} \quad &\|x - x_k\| \leq \Delta.
\end{aligned}$$

Adding now two auxiliary variables $t \in \mathbb{R}_+$ and $v \in \mathbb{R}^d$ with the constraints:

$$\frac{1}{x^\top \widehat{\mu}} = t, \quad v = tx,$$

the claim in the statement of the theorem now follows by a simple substitution to get

$$\begin{aligned}
\max \quad &\frac{1}{\sqrt{v^\top \widehat{\Sigma} v}} \\
\text{s.t.} \quad &v \in \mathbb{R}^d, \; t \in \mathbb{R}_+, \; \|v - tx_k\|_2 \leq \Delta t, \; v^\top \widehat{\mu} = 1.
\end{aligned}$$

Swapping the maximum operator to a minimum operator completes the proof. $\qquad\square$

## A.3 PROOFS OF SECTION 4

*Proof of Lemma 4.1.* From the definition of the set $\Theta(\{x_j\})$, we have

$$
\max\{r : \mathcal{E}_r \subseteq \Theta(\{x_j\})\} =
\begin{cases}
\max & r \\
\text{s.t.} & \sup_{\|u\|_2 \leq r} -(\widehat{\Sigma}^{\frac{1}{2}} u + \widehat{\mu})^\top x_j \leq 0 \qquad \forall j
\end{cases}
$$

$$
=
\begin{cases}
\max & r \\
\text{s.t.} & -\widehat{\mu}^\top x_j + \sup_{\|u\|_2 \leq r} -x_j^\top \widehat{\Sigma}^{\frac{1}{2}} u \leq 0 \qquad \forall j
\end{cases}
$$

$$
=
\begin{cases}
\max & r \\
\text{s.t.} & -\widehat{\mu}^\top x_j + r\|\widehat{\Sigma}^{\frac{1}{2}} x_j\|_2 \leq 0 \qquad \forall j,
\end{cases}
$$

where the last equality follows from the dual norm property. The proof now follows by finding the maximum value of $r$ so that the problem is feasible. □

## B  EXPERIMENTS

### B.1  EXPERIMENTAL DETAIL

**Real-world datasets**   Here, we provide more detail about the three real-world datasets we used. Source code can be found at `https://github.com/ngocbh/COPA`.

  i *German Credit* (Dua & Graff, 2017). The dataset contains the information (e.g. age, gender, financial status,...) of 1000 customers who took a loan from a bank. The classification task is to determine the risk (good or bad) of an individual. There is another version of this dataset regarding corrections of coding error (Groemping, 2019). We use the corrected version of this dataset as shifted data to capture the correction shift. The features we used in this dataset include 'duration', 'amount', 'personal_status_sex', and 'age'.

 ii *Small Bussiness Administration (SBA)* (Li et al., 2018). This data includes 2,102 observations with historical data of small business loan approvals from 1987 to 2014. We divide this dataset into two datasets (one is instances from 1989 - 2006 and one is instances from 2006 - 2014) to capture temporal shift. We use the following features: selected, 'Term', 'NoEmp', 'CreateJob', 'RetainedJob', 'UrbanRural', 'ChgOffPrinGr', 'GrAppv', 'SBA_Appv', 'New', 'RealEstate', 'Portion', 'Recession'.

iii *Student performance* (Cortez & Silva, 2008). This data includes the performance records of 649 students in two schools: Gabriel Pereira (GP) and Mousinho da Silveira (MS). The classification task is to determine if their final score is above average or not. We split this dataset into two sets in two schools to capture geospatial shift. The features we used are: 'age', 'Medu', 'Fedu', 'studytime', 'famsup', 'higher', 'internet', 'romantic', 'freetime', 'goout', 'health', 'absences', 'G1', 'G2'.

**Classifier**   Throughout this paper, we use a Logistic Regression for a linear classifier and a three-layer MLP with 20, 50, 20 nodes and ReLU activation in each consecutive layer as the nonlinear classifier. We use one-hot encoding for categorical features in the datasets to convert it to a vector of $[0, 1]$. We use min-max normalization to scale the numerical features to [0, 1]. We report the performance of the classifiers in three real-world datasets in Table 2

### B.2  ADDITIONAL EXPERIMENTS

**The impact of degree of distribution shift on validity of a plan.**   We provide an additional experiment in different covariance shift $\Sigma_g = (1 + \beta)A, A \succeq 0$. In this experiment, we choose $A$ as:

$$
A = \begin{pmatrix} 1 & -1 & 0 \\ -1 & 1 & 1 \\ 0 & 1 & 1 \end{pmatrix}.
$$

The matrix $A$ introduces both positive and negative correlations between the classifier's parameters. Other settings are set the same as the experiment in Section 5.1.

Table 2: Performance of the underlying classifiers.

| | Logistic Regression | | Neural Network | |
|---|---|---|---|---|
| | Accuracy | AUC | Accuracy | AUC |
| German | $0.71 \pm 0.01$ | $0.64 \pm 0.02$ | $0.68 \pm 0.02$ | $0.62 \pm 0.02$ |
| Shifted German | $0.71 \pm 0.01$ | $0.64 \pm 0.02$ | $0.68 \pm 0.02$ | $0.62 \pm 0.02$ |
| SBA | $0.71 \pm 0.02$ | $0.86 \pm 0.02$ | $0.96 \pm 0.02$ | $0.99 \pm 0.01$ |
| Shifted SBA | $0.87 \pm 0.01$ | $0.90 \pm 0.02$ | $0.97 \pm 0.01$ | $0.98 \pm 0.01$ |
| Student | $0.83 \pm 0.02$ | $0.91 \pm 0.02$ | $0.88 \pm 0.02$ | $0.95 \pm 0.01$ |
| Shifted Student | $0.87 \pm 0.03$ | $0.93 \pm 0.03$ | $0.90 \pm 0.03$ | $0.96 \pm 0.01$ |

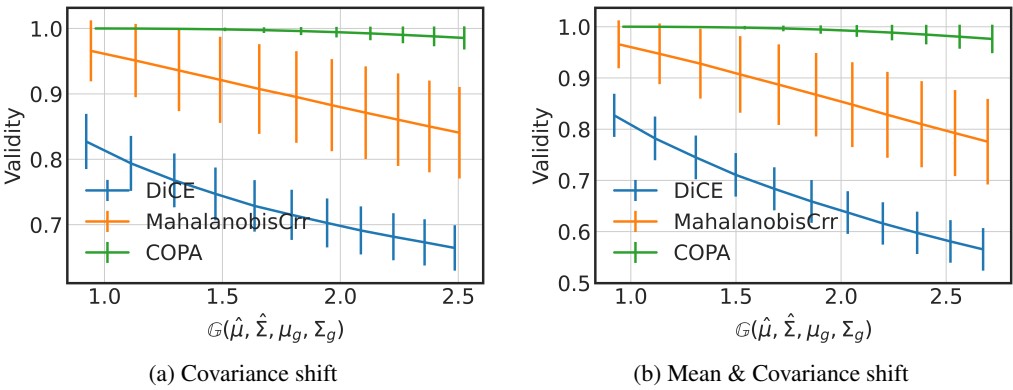

(a) Covariance shift          (b) Mean & Covariance shift

Figure 5: The impact of shift magnitudes on the validity of the plans obtained by three algorithms.

**Mahalanobis correction on real-world datasets.** In this experiment, we evaluate the Mahalanobis correction on different number of corrections $K$ and different perturbation limit $\Delta$. We set $\rho = 0.01, \epsilon = 0.1, K \in \{0, \ldots, J\}, J = 5, \Delta \in [0.05, 0.35]$. $(\widehat{\mu}, \widehat{\Sigma})$ is estimated using similar manner as in Section 5.2 of the main paper. The results in shown in Figure 6.

**Counterfactual explanations for real-world datasets.** To illustrate the use case of the counterfactual explanations, we provide some examples on the German dataset with $J = 3$ (Table 3) in which we consider the "personal status and sex" feature as immutable. Here, we can observe that three algorithms could provide diverse sets of counterfactuals that the users may prefer. However, by providing better empirical validity, the plans generated by MahalanobisCrr and COPA are more robust with distribution shift than DiCE (generated without considering the shift).

## C  EXTENSION TO NONLINEAR CLASSIFIERS

In the main paper, our analysis is based on the linearity in both features and model parameters. We now discuss two extensions of our COPA framework to the nonlinear settings.

### C.1  NONLINEARITY IN INPUT FEATURES

This section extends to any linear classifier $\mathcal{C}_\theta(x) = 1$ if $\theta^\top \phi(x) \geq 0$, and 0 otherwise, where $\phi : \mathcal{X} \to \mathbb{R}^d$ is a (possibly nonlinear) feature mapping that maps input features to a latent representation in a covariate space $\mathbb{R}^d$. Note that our bounds in Section 2 still hold in latent space $\mathbb{R}^d$: for a concrete example, Theorem 2.2 holds with $x_j$ being replaced by $\phi(x_j)$.

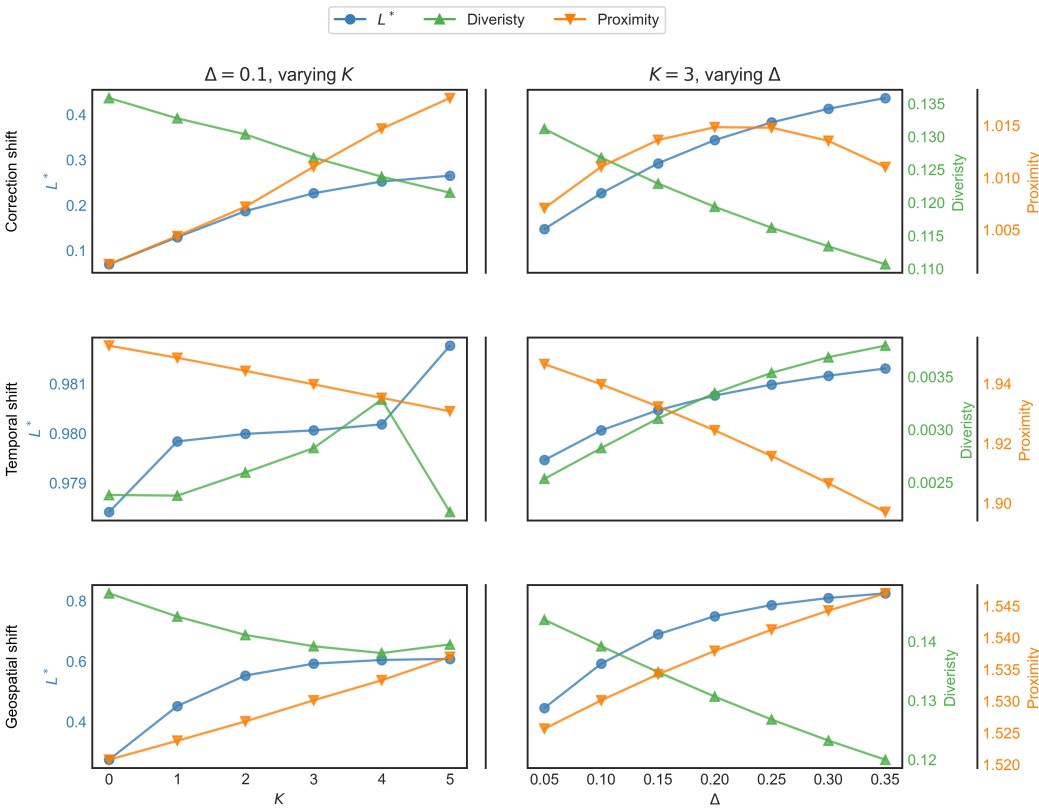

Figure 6: Evaluation of Mahalanobis correction on the real-world datasets. We fix $\Delta = 0.1$, evaluate the effect of the number of correction on the lower validity bound, diversity, and proximity (left column). We fix $K = 3$, evaluate the effect of the perturbation limit on the lower validity bound, diversity, and proximity (right column).

The COPA framework is also extendable to incorporate the feature map $\phi$. Assuming that $\phi$ is differentiable, the COPA framework solves the following optimization problem:

$$
\min_{x_1,\ldots,x_J} \quad \text{Proximity}(\{x_j\}, x_0) + \lambda_1 \text{Validity}(\{\phi(x_j)\}, \widehat{\mu}, \widehat{\Sigma}) - \lambda_2 \text{Diversity}(\{x_j\})
$$
$$
\text{s.t.} \quad \widehat{\mu}^\top x_j \geq \epsilon \qquad \forall j.
$$
(8)

The proximity and diversity are measured in the input space and the validity term is now measured in latent space instead. This optimization problem can be solved efficiently by a projected gradient descent algorithm similar to Section 4.

### C.2 NONLINEARITY IN MODEL'S PARAMETERS

Similar to the prior works (Ustun et al., 2019; Rawal & Lakkaraju, 2020; Upadhyay et al., 2021), our work can adapt to nonlinear classifiers $\mathcal{C}_{nl}$ using a local surrogate models such as LIME (Ribeiro et al., 2016). LIME (Ribeiro et al., 2016) is a popular technique for explaining predictions of black-box machine learning models. The main idea of LIME is to train a local surrogate model $\mathcal{C}_\theta^{x_0}$ on perturbed samples around a given input instance $x_0$ to approximate the local decision boundary of the black-box models. We thus model the uncertainty of parameters $\theta$ in the surrogate model $\mathcal{C}_\theta^{x_0}$ for $x_0$ instead of the parameters of $\mathcal{C}_{nl}$.

For the experiment, we first generate a local linear model $\mathcal{C}_\theta^{x_0}$ using LIME method with 5000 perturbed samples. We then choose $(\widehat{\mu}, \widehat{\Sigma}) = (\theta, 0.05I)$, where $I$ is identity matrix, to model the

| | Duration | Credit amount | Personal status | Age | $L^\star$ | Empirical Validity |
|---|---|---|---|---|---|---|
| Instance | 30.0 | 4249.0 | A94 | 28.0 | - | - |
| DiCE | 49.4 | 596.4 | - | 28.0 | 0.00 | 0.005 |
| | 72.0 | 4330.2 | - | 28.0 | | |
| | 59.7 | 13776.4 | - | 28.0 | | |
| MahalanobisCrr | 42.5 | 69.8 | - | 30.0 | 0.15 | 0.802 |
| | 40.8 | 1153.9 | - | 30.0 | | |
| | 27.4 | 10047.3 | - | 30.0 | | |
| COPA | 4.0 | 18424.0 | - | 28.0 | 0.11 | 0.797 |
| | 72.0 | 9410.3 | - | 28.0 | | |
| | 40.3 | 250.0 | - | 28.0 | | |
| Instance | 42.0 | 7174.0 | A92 | 30.0 | - | - |
| DiCE | 11.8 | 250.0 | - | 30.0 | 0.38 | 0.88 |
| | 4.0 | 7167.4 | - | 30.0 | | |
| | 13.4 | 13386.8 | - | 30.0 | | |
| MahalanobisCrr | 7.9 | 523.7 | - | 33.0 | 0.61 | 0.968 |
| | 3.6 | 5500.1 | - | 32.0 | | |
| | 9.1 | 12234.5 | - | 32.0 | | |
| COPA | 4.0 | 3884.7 | - | 30.0 | 0.88 | 1.000 |
| | 16.3 | 3280.8 | - | 30.0 | | |
| | 15.5 | 250.0 | - | 30.0 | | |
| Instance | 24.0 | 4526.0 | A93 | 74.0 | - | - |
| DiCE | 72.0 | 2165.7 | - | 74.0 | 0.00 | 0.080 |
| | 72.0 | 9907.4 | - | 74.0 | | |
| | 72.0 | 18424.0 | - | 74.0 | | |
| MahalanobisCrr | 62.1 | 1766.4 | - | 75.0 | 0.01 | 0.614 |
| | 55.6 | 8881.0 | - | 75.0 | | |
| | 48.8 | 16680.9 | - | 75.0 | | |
| COPA | 4.0 | 250.0 | - | 74.0 | 0.59 | 0.997 |
| | 44.8 | 3070.5 | - | 74.0 | | |
| | 4.0 | 18424.0 | - | 74.0 | | |

Table 3: Counterfactual examples on German dataset.

distributional uncertainty of the parameters. Similar to Section 5.2, we set Gelbrich radius $\rho$ is to 0.01, $J = 5$, $K = 3$.

Table 4: Performance of competing algorithms on nonlinear classifiers. The current validity is the validity of counterfactual plan with respect to the current nonlinear classifier $\mathcal{C}_{nl}$ (i.e., the fraction of instances that the generated counterfactual plan is feasible).

| Dataset | Method | Proximity | Diversity | $L^\star$ | Empirical Validity | Current Validity |
|---|---|---|---|---|---|---|
| Correction | DiCE | $0.515 \pm 0.204$ | $0.043 \pm 0.037$ | $0.005 \pm 0.041$ | $0.414 \pm 0.238$ | **0.990** |
| | MahalanobisCrr | $0.595 \pm 0.210$ | $0.035 \pm 0.035$ | $0.021 \pm 0.058$ | $0.409 \pm 0.313$ | 0.670 |
| | COPA ($\lambda_1 = 0.1; \lambda_2 = 1.0$) | **0.219** $\pm 0.183$ | $0.001 \pm 0.011$ | **0.065** $\pm 0.088$ | **0.556** $\pm 0.331$ | 0.560 |
| | COPA ($\lambda_1 = 0.1; \lambda_2 = 2.0$) | $0.432 \pm 0.403$ | $0.100 \pm 0.116$ | $0.049 \pm 0.093$ | $0.301 \pm 0.341$ | 0.270 |
| | COPA ($\lambda_1 = 0.2; \lambda_2 = 2.0$) | $0.673 \pm 0.314$ | **0.162** $\pm 0.084$ | $0.038 \pm 0.097$ | $0.125 \pm 0.186$ | 0.040 |
| Temporal | DiCE | $1.573 \pm 0.451$ | $0.107 \pm 0.071$ | $0.637 \pm 0.350$ | $0.852 \pm 0.270$ | **1.000** |
| | MahalanobisCrr | $1.567 \pm 0.449$ | $0.099 \pm 0.070$ | $0.868 \pm 0.118$ | $0.987 \pm 0.076$ | **1.000** |
| | COPA ($\lambda_1 = 0.1; \lambda_2 = 1.0$) | **1.388** $\pm 0.540$ | $0.002 \pm 0.008$ | $0.981 \pm 0.014$ | **1.000** $\pm 0.000$ | **1.000** |
| | COPA ($\lambda_1 = 0.1; \lambda_2 = 2.0$) | $1.534 \pm 0.408$ | **0.247** $\pm 0.043$ | $0.976 \pm 0.012$ | **1.000** $\pm 0.000$ | **1.000** |
| | COPA ($\lambda_1 = 0.2; \lambda_2 = 2.0$) | $1.447 \pm 0.340$ | $0.118 \pm 0.072$ | **0.990** $\pm 0.004$ | **1.000** $\pm 0.000$ | **1.000** |
| Geospatial | DiCE | $1.576 \pm 0.349$ | $0.175 \pm 0.070$ | $0.022 \pm 0.046$ | $0.328 \pm 0.303$ | **1.000** |
| | MahalanobisCrr | $1.594 \pm 0.349$ | $0.169 \pm 0.071$ | $0.113 \pm 0.084$ | $0.689 \pm 0.280$ | **1.000** |
| | COPA ($\lambda_1 = 0.1; \lambda_2 = 1.0$) | **1.342** $\pm 0.367$ | $0.000 \pm 0.000$ | $0.011 \pm 0.007$ | $0.384 \pm 0.310$ | 0.710 |
| | COPA ($\lambda_1 = 0.1; \lambda_2 = 2.0$) | $1.552 \pm 0.292$ | $0.243 \pm 0.039$ | $0.010 \pm 0.024$ | $0.168 \pm 0.210$ | 0.750 |
| | COPA ($\lambda_1 = 0.2; \lambda_2 = 2.0$) | $1.637 \pm 0.284$ | **0.287** $\pm 0.017$ | **0.164** $\pm 0.066$ | $0.679 \pm 0.274$ | **1.000** |

We report the performance of three algorithms on the MLP classifier in the real-world datasets in Table 4. The result is promising since the proposed COPA can increase the empirical validity

significantly. However, the infidelity of LIME could lead to invalid counterfactual explanations, represented by a lower current validity value. The low current validity is also observed in the literature, see Upadhyay et al. (2021). For further investigation, one can use another local surrogate model that provides a better approximation of the decision boundary (e.g., BayLIME (Zhao et al., 2020)). Another direction is to use a mixture linear regression model to approximate the decision boundary as in Guo et al. (2018). However, advocating for the mixture of linear models requires further analysis.

