# OpenReview forum: "Counterfactual Plans under Distributional Ambiguity"
_ICLR.cc/2022/Conference — ICLR 2022 Poster_

### Official Review · Reviewer_qAnf · 2021-11-02

**Correctness:** 4
**Technical Novelty And Significance:** 4
**Empirical Novelty And Significance:** 3
**Recommendation:** 8
**Confidence:** 4

**Main Review:**

Very interesting paper on a highly relevant subject that rightfully draws a lot of attention these days. It is an application of explainable AI in terms of how a user should change their ‘features’ in order for them to be classified differently by a complex system. Typical examples include loan/student/job applications, where a user may want to know why they got rejected and what they would need to change in order to be successful next time, which is known as the ‘counterfactual plan’. Crucial in this problem is that different users may have different preferences/possibilities to change, and therefore a single advice / explanation to all users (even with the same covariates) may not be the best strategy, which is why it makes sense to aim for a collection of plans, allowing a user to decide for themselves which course of action they may prefer. This problem is compounded by the fact that at a later point in time certain parameters in the classification model may have changed, which may invalidate the original counterfactual plan.

The approach to tackle this problem in the paper is solid and well described. The rationale to model parameter uncertainty by a distribution characterised by the first two moments leads to meaningful and useful bounds on the probability of success (validity) of a given plan (Thm2.2+3), although in practice it will be rather difficult to assess this for real. Also it may not capture realistic changes like exclusion of certain (politically sensitive) features from the model, or inclusion of new features.
The solution. to stat from an optimal plan under fixed parameters and then to modify/correct this based on the aforementioned distribution to improve the probability of validty is also intuitive and effective.
The desiderata for the collection of counterfactual plans also makes intuitive sense: proximity as smaller changes are likely to be more doable, diversity to give users true alternatives that suit their needs/options, and validity because it is nice the plan is actually likely to be effective.
Numerical experiments in section 5 support the claims on the bounds and show the COPA framework as a whole is effective in finding plans that satisfy the target criteria, although the real-world experiments in 5.2 are on the overly short/compressed side. As a result it is difficult to truly asses the usefulness of the final output in practice.

Nevertheless, a few remarks / questions remain.

1. The general setup is that of a linear classifier, however the characterisation at the bottom of p2 is much more restrictive. Usually a classifier is considered linear if it is linear in terms of the parameters, and the input variables are typically transformed into a feature vector \phi(x) consisting of a collection of basis functions based on the input. But these feature vectors themselves can then be highly nonlinear in x, and are almost always also highly dependent. It allows for complex decision boundaries in a classifier that reduce to a linear decision hyperplane in some high dimensional space. This way for example a classifier can take into account that a certain age group is likely to be more successful rather than ‘the older the better’. But that also means that it is not possible for a user to directly or independently to intervene on each of these features in the classifier: they can only interact with the input x_i … each of which can be present in many features per parameter of the classifier.This is why it is often so difficult to give good advice.

If I understand correctly the current approach effectively assumes the classifier is linear in *both* variables and parameters … which is not realistic in most real-world classifiers, apart from some overly simplistic systems. This would greatly limit the applicability of the proposed approach in practice.

2. The ‘diversity’ metric in eq.(5) is a useful target in itself, but I am not sure the det(K) factor does what is needed by a user to make an optimal choice. I think it eliminates plans that are linearly dependent (as then det(K) will be zero), but it is possible that a user may need to modify either x_1 or x_2 by a significant margin to ensure 95% probability of success (depending on the parameter shift distribution), but only by a relatively small amount if x_1 and x_2 can be modified simultaneously. Conversely a user may prefer to only alter one aspect rather than try to modify two at the same time  So then there are 3 reasonable plans involving two input variables for a user to choose from, but this would not be an output solution as it would imply a lack of diversity, right?

3. The paper would benefit from an actual example of a collection of counterfactual plans as generated by the COPA framework for a given case, both without and with parameter shift, as well as how two users with the same features but different preferences would use this to choose for different actions.

Minor comments:
p1,mid “in practice …’ => explain this corresponds to an individual/personal loss function / cost matrix that can differ per user … and later show how the output of your algorithm can be used differently by two users in with the same initial features but different preferences

p2, Gen.setup: counterfactual plan stated as ‘increase relative to current value’ or ‘get at least this absolute value’?
also: typo ‘explainaition’
p3,Def1: calling it ‘joint feasibility’ suggests it is more than what it is (simply ‘all plans should be feasible’)
p4,Thm2: discuss how to interpret the role of z_j in Thm 2.2
-
p5,3.2 ‘Correction procedure’, ‘if we can adjust K out of J actions’ => why would there be a restriction on the number of adjustments? You simply give an updated advice / plan, and each of the suggestions in there should take the adjustment into account, so there is no reason to take K < J.
p6, Proximity, eq(4): it would be good to discuss the role of the cost function c and explain it can be used to capture ‘general indication of the ease of change’ of a specific variable, in order to avoid suggesting people change gender or height.
p7, eq(6): it seems a ‘diversity’ got mixed up with a ‘validity’ component here
p9, 5.2+closing remark: this is too compact; also it would be good to assess how useful users perceive the resulting output to be


**Summary Of The Paper:**

The paper considers the problem of how to provide a collection of counterfactual explanations for a binary linear classifier such that an informed choice can be made on how best to actualise a new input based on individual preferences such that a different classification is likely to be made, even under potentially changing model parameters. It introduces the Counterfactual Plan under Ambiguity (COPA) framework, consisting of a probabilistic validity assessment of a predetermined plan under a given parameter distribution of the classifier, followed by a correction of the predetermined plan to improve validity under that distribution. These are combined in the COPA framework to produce a plan that optimises a weighted combination of the desiderata ‘proximity’, ‘diversity’ and ‘validity’. Experiments show the resulting output satisfies the derived upper and lower bounds on the validity, with good performance on the desiderata.


**Summary Of The Review:**

Solid paper, important problem, interesting solution. Some caveats (in particular the assumption the classifier is linear in the input variables seems unrealistic for most real-world classifiers, potentially severely limiting the applicability of the COPA framework in practice), but overall clear accept.

---

> ### Author Response · Authors · 2021-11-22
> **Response to Reviewer qAnf [Part 1/3]**
>
> Thank you for your thoughtful review and valuable feedback. We appreciate your interest in our work. Below we address the reviewer’s questions:
>
> **(1) General setup of linear classifiers**: Thank you so much for pointing out an excellent point. Throughout the paper, our analysis is indeed based on the linearity in both variables and parameters. However, our work could extend to any linear classifier $\mathcal{C}_\theta(x) = 1$ if $\theta^\top \phi(x) \ge 0$, and $0$ otherwise, where $\phi: \mathcal{X} \rightarrow \mathbb{R}^d$ is a (possibly nonlinear) feature mapping that maps input features to a latent representation in a covariate space $\mathbb{R}^d$. Note that our bounds in Section 2 still hold in latent space $\mathbb{R}^d$: for a concrete example, Theorem 2.2 holds with $x_j$ being replaced by $\phi(x_j)$.
>
>  The COPA framework is also extendable to incorporate the feature map $\phi$. Assuming that $\phi$ is differentiable, the COPA framework solves the following optimization problem:
>
> $$
> \begin{array}{cl}
>         \mathrm{min}_{x_1, \ldots, x_J} & \mathrm{Proximity}(\{x_j\}, x_0) + \lambda_1  \mathrm{Validity}(\{\phi(x_j)\}, \hat{\mu}, \hat{\Sigma} ) - \lambda_2 \mathrm{Diversity}(\{x_j\}) \\\\
>         \mathrm{s.t.} & \hat{\mu}^\top x_j \ge \epsilon \qquad \forall j
> \end{array}
> $$
>
> The proximity and diversity are now measured in the input space and the validity term is measured in latent space.
> This optimization problem can be solved efficiently by projected gradient descent similar to Section 4. We also added this discussion in Appendix C.1 of the manuscript.
>
> **(2) Diversity metric**: The determinantal point process (DPP) has been widely applied in many problems in machine learning to measure the diversity or sample a diverse subset from a given set (see Kulesza et al. [13], Mothilal et al. [12]). We argue that the diversity metric is not affected by the linear dependence in $\\{x_j\\}$. A counterexample that we could provide is $x_0 = [1, 0], x_2 = [0, 1], x_3 = [1, 1]$.  We then have a correlation kernel
> $
> K = \begin{pmatrix}
>     1 & 0.414 & 0.5,\\\\
>     0.414 & 1 & 0.5, \\\\
>     0.5 & 0.5 & 1
>     \end{pmatrix}.
> $
> This set turns out to have $\mathrm{Diversity}(\\{x_j\\}) = det(K) = 0.53553$. A case that has
> $\mathrm{Diversity}(\\{x_j\\}) = 0$ is when two points overlapped. Indeed, if there are two points overlapped in the set, says $x_i$ and $x_j$, the distance from any other point to $x_i$ and $x_j$ is equal. So the column $K_i$ is the same as the column $K_j$. That leads to the determinant of $K$ is zero.
>
> **(3) Explanations for real-world datasets.**: Thank you for the suggestion. We provide an example of three plans generated on the German dataset with $J=3$. Here we consider the `personal status and sex` feature as immutable. We can observe that three algorithms could provide a diverse set of counterfactuals that the users may prefer. However, by providing better empirical validity, the plans generated by MahalanobisCrr and COPA are expected to be more robust with distribution shift than DiCE (generated without considering the shift). Due to space constraints, we have included those examples in Appendix B.2.
>
> |                | Duration | Credit amount | Personal status | Age  | $L^\star$ | Empirical Validity |
> |----------------|----------|---------------|-----------------|------|---------|--------------------|
> | Instance       | 42.0     | 7174.0        | A92             | 30.0 | -       | -                  |
> | DiCE           | 11.8     | 250.0         | -               | 30.0 | 0.38    | 0.88               |
> |                | 4.0      | 7167.4        | -               | 30.0 |         |                    |
> |                | 13.4     | 13386.8       | -               | 30.0 |         |                    |
> | MahalanobisCrr | 7.9      | 523.7         | -               | 33.0 | 0.61    | 0.968              |
> |                | 3.6      | 5500.1        | -               | 32.0 |         |                    |
> |                | 9.1      | 12234.5       | -               | 32.0 |         |                    |
> | COPA           | 4.0      | 3884.7        | -               | 30.0 | 0.88    | 1.000              |
> |                | 16.3     | 3280.8        | -               | 30.0 |         |                    |
> |                | 15.5     | 250.0         | -               | 30.0 |         |                    |

---

> > ### Author Response · Authors · 2021-11-22
> > **Response to Reviewer qAnf [Part 2/3]**
> >
> > **(4) Minor**: Thank you for many constructive comments. We have updated the manuscript according to your comments. We address the remaining question below:
> > * p5, 3.2 “Correction procedure”: The reason for correcting $K$ out of $J$ counterfactuals: As demonstrated in the previous works (Rawal et al. [1], Upadhyay et al. [2]), introducing the robustness requires sacrificing proximity (cost of changes) of the counterfactuals. Therefore, correcting only $K$ out of $J$ counterfactuals allows the flexibility to balance the trade-off between the cost and robustness. Concretely, assuming that we have a limited budget of sacrificing the proximity, it is reasonable to share it for $K$ counterfactuals that improve the empirical validity most rather than sharing for all $J$ counterfactuals (see the Geometric intuition in Section 3.2 and the trade-off experiment of Mahalanobis correction in Section B.2 in the appendix).
> >
> > **(5) Extension to any nonlinear classifiers**: Similar to some previous works (Ustun et al [3], Rawal et al. [1], Upadhyay et al. [2]), our work can extend to nonlinear classifiers by using a local linear model that approximates the decision boundary around a given input instance (e.g. LIME [6]). To evaluate this approach, we have conducted an experiment with a nonlinear classifier (three-layer MLP) in three real-world datasets. We have included this experiment in Appendix C.2 of the paper. We report the results below:
> >
> > |Dataset        |Method                                       |Proximity               |Diversity                |$L^*$                |Empirical Validity                 |Current Validty     |
> > |----------|-----------------------------------------|---------------------|---------------------|---------------------|---------------------|---------|
> > |Correction|DiCE                                     |0.515 $\pm$ 0.204    |0.043 $\pm$ 0.037    |0.005 $\pm$ 0.041    |0.414 $\pm$ 0.238    |**0.990**|
> > |          |MahalanobisCrr                           |0.595 $\pm$ 0.210    |0.035 $\pm$ 0.035    |0.021 $\pm$ 0.058    |0.409 $\pm$ 0.313    |0.670    |
> > |          |COPA ($\lambda_1 = 0.1; \lambda_2 = 1.0$)|**0.219** $\pm$ 0.183|0.001 $\pm$ 0.011    |**0.065** $\pm$ 0.088|**0.556** $\pm$ 0.331|0.560    |
> > |          |COPA ($\lambda_1 = 0.1; \lambda_2 = 2.0$)|0.432 $\pm$ 0.403    |0.100 $\pm$ 0.116    |0.049 $\pm$ 0.093    |0.301 $\pm$ 0.341    |0.270    |
> > |          |COPA ($\lambda_1 = 0.2; \lambda_2 = 2.0$)|0.673 $\pm$ 0.314    |**0.162** $\pm$ 0.084|0.038 $\pm$ 0.097    |0.125 $\pm$ 0.186    |0.040    |
> > |Temporal  |DiCE                                     |1.573 $\pm$ 0.451    |0.107 $\pm$ 0.071    |0.637 $\pm$ 0.350    |0.852 $\pm$ 0.270    |**1.000**|
> > |          |MahalanobisCrr                           |1.567 $\pm$ 0.449    |0.099 $\pm$ 0.070    |0.868 $\pm$ 0.118    |0.987 $\pm$ 0.076    |**1.000**|
> > |          |COPA ($\lambda_1 = 0.1; \lambda_2 = 1.0$)|**1.388** $\pm$ 0.540|0.002 $\pm$ 0.008    |0.981 $\pm$ 0.014    |**1.000** $\pm$ 0.000|**1.000**|
> > |          |COPA ($\lambda_1 = 0.1; \lambda_2 = 2.0$)|1.534 $\pm$ 0.408    |**0.247** $\pm$ 0.043|0.976 $\pm$ 0.012    |**1.000** $\pm$ 0.000|**1.000**|
> > |          |COPA ($\lambda_1 = 0.2; \lambda_2 = 2.0$)|1.447 $\pm$ 0.340    |0.118 $\pm$ 0.072    |**0.990** $\pm$ 0.004|**1.000** $\pm$ 0.000|**1.000**|
> > |Geospatial|DiCE                                     |1.576 $\pm$ 0.349    |0.175 $\pm$ 0.070    |0.022 $\pm$ 0.046    |0.328 $\pm$ 0.303    |**1.000**|
> > |          |MahalanobisCrr                           |1.594 $\pm$ 0.349    |0.169 $\pm$ 0.071    |0.113 $\pm$ 0.084    |**0.689** $\pm$ 0.280|**1.000**|
> > |          |COPA ($\lambda_1 = 0.1; \lambda_2 = 1.0$)|**1.342** $\pm$ 0.367|0.000 $\pm$ 0.000    |0.011 $\pm$ 0.007    |0.384 $\pm$ 0.310    |0.710    |
> > |          |COPA ($\lambda_1 = 0.1; \lambda_2 = 2.0$)|1.552 $\pm$ 0.292    |0.243 $\pm$ 0.039    |0.010 $\pm$ 0.024    |0.168 $\pm$ 0.210    |0.750    |
> > |          |COPA ($\lambda_1 = 0.2; \lambda_2 = 2.0$)|1.637 $\pm$ 0.284    |**0.287** $\pm$ 0.017|**0.164** $\pm$ 0.066|0.679 $\pm$ 0.274    |**1.000**|
> >
> >
> > The result is promising since the proposed methods can increase the empirical validity significantly. However, the infidelity of LIME could lead to invalid counterfactual explanations, represented by a lower current validity value. The low current validity is also observed in the literature, see Upadhyay et al. (2021). For further investigation, one can use another local surrogate model that provides a better approximation of the decision boundary (e.g. BayLIME [8]). Another direction that should be considered is to use a mixture linear regression model to approximate the decision boundary as in Guo et al [9].  However, advocating for the mixture of linear models requires further analysis.

---

> > > ### Author Response · Authors · 2021-11-22
> > > **Response to Reviewer qAnf [Part 3/3]**
> > >
> > > We hope that we have addressed all your concerns adequately. We have also revised the manuscript according to your comments. Please let us know if we can provide any further details and/or clarifications.
> > >
> > > *References*:
> > > - [1] Rawal, Kaivalya, Ece Kamar, and Himabindu Lakkaraju. "Can I Still Trust You?: Understanding the Impact of Distribution Shifts on Algorithmic Recourses." arXiv:2012.11788 (2020).
> > > - [2] Upadhyay, Sohini, Shalmali Joshi, and Himabindu Lakkaraju. "Towards Robust and Reliable Algorithmic Recourse." arXiv preprint arXiv:2102.13620 (2021).
> > > - [3] Ustun, Berk, Alexander Spangher, and Yang Liu. "Actionable recourse in linear classification." Proceedings of the Conference on Fairness, Accountability, and Transparency. 2019.
> > > - [4] Karimi, Amir-Hossein, Bernhard Schölkopf, and Isabel Valera. "Algorithmic recourse: from counterfactual explanations to interventions." Proceedings of the 2021 ACM Conference on Fairness, Accountability, and Transparency. 2021.
> > > - [5] Karimi, Amir-Hossein, et al. "Algorithmic recourse under imperfect causal knowledge: a probabilistic approach." arXiv preprint arXiv:2006.06831 (2020).
> > > - [6] Ribeiro, Marco Tulio, Sameer Singh, and Carlos Guestrin. "" Why should i trust you?" Explaining the predictions of any classifier." Proceedings of the 22nd ACM SIGKDD international conference on knowledge discovery and data mining. 2016.
> > > - [7] Chris Russell. Efficient search for diverse coherent explanations. In Proceedings of the Conference on Fairness, Accountability, and Transparency, pp. 20–28, 2019.
> > > - [8] Zhao, Xingyu, et al. "Baylime: Bayesian local interpretable model-agnostic explanations." arXiv preprint arXiv:2012.03058 (2020).
> > > - [9] Guo, Wenbo, et al. "Lemna: Explaining deep learning based security applications." Proceedings of the 2018 ACM SIGSAC Conference on Computer and Communications Security. 2018.
> > > - [10] Mahajan, Divyat, Chenhao Tan, and Amit Sharma. "Preserving causal constraints in counterfactual explanations for machine learning classifiers." arXiv preprint arXiv:1912.03277 (2019).
> > > - [11] Samoilescu, Robert-Florian, Arnaud Van Looveren, and Janis Klaise. "Model-agnostic and Scalable Counterfactual Explanations via Reinforcement Learning." arXiv preprint arXiv:2106.02597 (2021).
> > > - [12] Mothilal, Ramaravind K., Amit Sharma, and Chenhao Tan. "Explaining machine learning classifiers through diverse counterfactual explanations." Proceedings of the 2020 Conference on Fairness, Accountability, and Transparency. 2020.
> > > - [13] Kulesza, Alex, and Ben Taskar. "Determinantal point processes for machine learning." arXiv preprint arXiv:1207.6083 (2012).

---

> > > > ### Author Response · Authors · 2021-11-30
> > > > **Response to Reviewer qAnf - For any further questions**
> > > >
> > > > Dear reviewer,
> > > >
> > > > We thank the reviewer again for your insightful reviews and thoughtful suggestions. We have done our best to address the questions raised in your review. The discussion period is coming to a close within a day and we remain open to discussing any remaining concerns you may have until the very end. If there are any remaining questions or concerns, please do not hesitate to ask.
> > > >
> > > > Thank you for taking the time to evaluate our work and our responses.
> > > >
> > > > Authors

---

### Official Review · Reviewer_pA7G · 2021-11-02

**Correctness:** 4
**Technical Novelty And Significance:** 3
**Empirical Novelty And Significance:** 3
**Recommendation:** 6
**Confidence:** 3

**Main Review:**

Overall, the studied problem is interesting, the writing is clear, and the results seem sound (but I did not check the proofs). I have a question about originality. What are the differences of the proposed method, compared to the previous work, such as Rawal et al. (2020), Upadhyay et al. (2021), Ustun et al., (2019), and Karimi et al. (2020), except that the previous ones consider a single counterfactual setting?

Moreover, in this paper, the authors only study the joint feasibility. In my view, it is also very important to study the feasibility of each element in the plan, because it can help us understand the usefulness of each element and thus make changes in a principled way. Can the authors explain a bit about it?

One more question: How difficult can these results be extended to the nonlinear scenario, since in most cases, the predictive model should be nonlinear?

Minor issue: The conclusion section is missing. It would be better to have it.

Post-rebuttal:

Thank you for the feedback. Your responses well addressed my concerns. Please add them in the revised version. I have increased the score.

**Summary Of The Paper:**

This paper studies counterfactual plans under model uncertainty, that is, the parameters of the predictive model shift when the new data come. In particular, the authors focus on a linear classification setting, based on which, they provide a lower and upper bound on the probability of joint feasibility of a given counterfactual plan, a correction method to improve the lower bound, and a framework to construct a counterfactual plan that satisfies certain optimality.


**Summary Of The Review:**

Interesting problem and solid results, but the assumptions (linear classifications) are very strong

---

> ### Author Response · Authors · 2021-11-22
> **Response to Reviewer pA7G [Part 1/3]**
>
> Thank you for your insightful comments. We appreciate your interest in our paper. Below we address your questions:
>
> **(1) Originality/Comparison to previous works**: In order to clarify the originality and novelty of this paper, we would like to highlight the key differences compared to previous works as follows:
> Ustun et al. [3] consider the problem of algorithmic recourse for linear classifiers as an optimization problem and use integer programming tools to minimize the cost of performing recourse actions efficiently. In particular, they define a set of actions (a.k.a. flipsets) that should be performed by only considering the feature values that already exist in the data so that the generated recourses are "actionable". A similar approach using mixed integer programming to generate a diverse set of counterfactuals is proposed in Russel [7]. Karimi et al. [4] [5] instead suggest considering the actions as interventions in a causal model so that taking an action is not feature independent but affects the downstream variables in the structural causal model (SCM).
>
> However, one limitation of the above works is only considering the feasibility of the generated counterfactuals with respect to the current classifier. It thus is vulnerable to the model updates that occurred due to distributional shift subject to the arrival of new data. Rawal et al. [1] then put this problem in the investigation. The extensive empirical analysis has demonstrated that even the state-of-the-art recourse algorithms are not robust to the model updates and pointed out three major distributional shifts related to correction, temporal, and geospatial shift. Upadhyay et al. [2] later consider the problem of generating counterfactuals that are directly robust to a small perturbation in the model’s parameters. They leverage adversarial training to optimize a minimax objective which is a linear combination of the counterfactual validity and its cost.
>
> On the other hand, our work instead focuses on the multiple counterfactual explanations under the uncertainty of the distributional shift. Beyond the extension to multiple counterfactuals case, our work has some major advantages as follows:
> * Upadhyay et al. [2] only consider an additive model shift for linear model parameters in which perturbations are restricted within a norm-ball or a rectangle centered by the parameters of the current classifier. However, in machine learning, not only input features but also model parameters have a strong correlation to each other. Thus, only using additive perturbations of model shift might not capture all types of shift. Furthermore, they treat all perturbation points in the norm-ball equally important, so that the generated counterfactuals might be overly conservative and hedge against a pathological parameter in the uncertainty set. Our approach instead considers the distributional shift under the known first and second moments that provides a more flexible setting for the shift of the model’s parameters since correlations between the model’s parameters are considered.
> * By defining an uncertainty set for distribution shift of the parameters, we propose a diagnostic tool to provide the worst-case assessment of probabilistic validity for a given counterfactual plan with respect to some parameter distributions of the classifier. This quantification inspires us to propose a correction that aims to increase the validity of the predetermined plan.
>
>
>
> **(2) Joint feasibility**: The main advantage of multiple counterfactual explanations is to provide different possibilities so that the users with the same features could choose different explanations that fit best with their preferences (usually cannot infer from data). Since we do not know the preferences of the users, we would like to generate a plan having the highest probability that all counterfactuals are valid in the future rather than producing a plan with mixed high and low validity (some counterfactuals have high empirical validity and some have low empirical validity). That is the reason driving us to consider joint feasibility for a counterfactual plan.
>
> However, our diagnostic tool can be easily applied to analyze the feasibility of each counterfactual independently by setting $J=1$ so that each counterfactual could be analyzed and corrected independently. This is an interesting point of view but needs to be put into further investigation. We will consider this as one of the directions for future works.

---

> > ### Author Response · Authors · 2021-11-22
> > **Response to Reviewer pA7G [Part 2/3]**
> >
> > **(3) Extension to nonlinear classifiers**: Similar to some previous works (Ustun et al [3], Rawal et al. [1], Upadhyay et al. [2]), our work can extend to nonlinear classifiers by using a local linear model that approximates the decision boundary around a given input instance (e.g. LIME [6]). To evaluate this approach, we have conducted an experiment with a nonlinear classifier (three-layer MLP) in three real-world datasets. We have included this experiment in Appendix C.2 of the paper. We report the table below:
> >
> > |Dataset        |Method                                       |Proximity               |Diversity                |$L^*$                |Empirical Validity                 |Current Validty     |
> > |----------|-----------------------------------------|---------------------|---------------------|---------------------|---------------------|---------|
> > |Correction|DiCE                                     |0.515 $\pm$ 0.204    |0.043 $\pm$ 0.037    |0.005 $\pm$ 0.041    |0.414 $\pm$ 0.238    |**0.990**|
> > |          |MahalanobisCrr                           |0.595 $\pm$ 0.210    |0.035 $\pm$ 0.035    |0.021 $\pm$ 0.058    |0.409 $\pm$ 0.313    |0.670    |
> > |          |COPA ($\lambda_1 = 0.1; \lambda_2 = 1.0$)|**0.219** $\pm$ 0.183|0.001 $\pm$ 0.011    |**0.065** $\pm$ 0.088|**0.556** $\pm$ 0.331|0.560    |
> > |          |COPA ($\lambda_1 = 0.1; \lambda_2 = 2.0$)|0.432 $\pm$ 0.403    |0.100 $\pm$ 0.116    |0.049 $\pm$ 0.093    |0.301 $\pm$ 0.341    |0.270    |
> > |          |COPA ($\lambda_1 = 0.2; \lambda_2 = 2.0$)|0.673 $\pm$ 0.314    |**0.162** $\pm$ 0.084|0.038 $\pm$ 0.097    |0.125 $\pm$ 0.186    |0.040    |
> > |Temporal  |DiCE                                     |1.573 $\pm$ 0.451    |0.107 $\pm$ 0.071    |0.637 $\pm$ 0.350    |0.852 $\pm$ 0.270    |**1.000**|
> > |          |MahalanobisCrr                           |1.567 $\pm$ 0.449    |0.099 $\pm$ 0.070    |0.868 $\pm$ 0.118    |0.987 $\pm$ 0.076    |**1.000**|
> > |          |COPA ($\lambda_1 = 0.1; \lambda_2 = 1.0$)|**1.388** $\pm$ 0.540|0.002 $\pm$ 0.008    |0.981 $\pm$ 0.014    |**1.000** $\pm$ 0.000|**1.000**|
> > |          |COPA ($\lambda_1 = 0.1; \lambda_2 = 2.0$)|1.534 $\pm$ 0.408    |**0.247** $\pm$ 0.043|0.976 $\pm$ 0.012    |**1.000** $\pm$ 0.000|**1.000**|
> > |          |COPA ($\lambda_1 = 0.2; \lambda_2 = 2.0$)|1.447 $\pm$ 0.340    |0.118 $\pm$ 0.072    |**0.990** $\pm$ 0.004|**1.000** $\pm$ 0.000|**1.000**|
> > |Geospatial|DiCE                                     |1.576 $\pm$ 0.349    |0.175 $\pm$ 0.070    |0.022 $\pm$ 0.046    |0.328 $\pm$ 0.303    |**1.000**|
> > |          |MahalanobisCrr                           |1.594 $\pm$ 0.349    |0.169 $\pm$ 0.071    |0.113 $\pm$ 0.084    |**0.689** $\pm$ 0.280|**1.000**|
> > |          |COPA ($\lambda_1 = 0.1; \lambda_2 = 1.0$)|**1.342** $\pm$ 0.367|0.000 $\pm$ 0.000    |0.011 $\pm$ 0.007    |0.384 $\pm$ 0.310    |0.710    |
> > |          |COPA ($\lambda_1 = 0.1; \lambda_2 = 2.0$)|1.552 $\pm$ 0.292    |0.243 $\pm$ 0.039    |0.010 $\pm$ 0.024    |0.168 $\pm$ 0.210    |0.750    |
> > |          |COPA ($\lambda_1 = 0.2; \lambda_2 = 2.0$)|1.637 $\pm$ 0.284    |**0.287** $\pm$ 0.017|**0.164** $\pm$ 0.066|0.679 $\pm$ 0.274    |**1.000**|
> >
> >
> >
> > The result is promising since the proposed methods can increase the empirical validity significantly. However, the infidelity of LIME could lead to invalid counterfactual explanations, represented by a lower current validity value. The low current validity is also observed in the literature, see Upadhyay et al. (2021). For further investigation, one can use another local surrogate model that provides a better approximation of the decision boundary (e.g. BayLIME [8]). Another direction that should be considered is to use a mixture linear regression model to approximate the decision boundary as in Guo et al [9].  However, advocating for the mixture of linear models requires further analysis.

---

> > > ### Author Response · Authors · 2021-11-22
> > > **Response to Reviewer pA7G [Part 3/3]**
> > >
> > > We hope that we have addressed all your concerns adequately. Please let us know if we can provide any further details and/or clarifications.
> > >
> > > *References*:
> > > - [1] Rawal, Kaivalya, Ece Kamar, and Himabindu Lakkaraju. "Can I Still Trust You?: Understanding the Impact of Distribution Shifts on Algorithmic Recourses." arXiv:2012.11788 (2020).
> > > - [2] Upadhyay, Sohini, Shalmali Joshi, and Himabindu Lakkaraju. "Towards Robust and Reliable Algorithmic Recourse." arXiv preprint arXiv:2102.13620 (2021).
> > > - [3] Ustun, Berk, Alexander Spangher, and Yang Liu. "Actionable recourse in linear classification." Proceedings of the Conference on Fairness, Accountability, and Transparency. 2019.
> > > - [4] Karimi, Amir-Hossein, Bernhard Schölkopf, and Isabel Valera. "Algorithmic recourse: from counterfactual explanations to interventions." Proceedings of the 2021 ACM Conference on Fairness, Accountability, and Transparency. 2021.
> > > - [5] Karimi, Amir-Hossein, et al. "Algorithmic recourse under imperfect causal knowledge: a probabilistic approach." arXiv preprint arXiv:2006.06831 (2020).
> > > - [6] Ribeiro, Marco Tulio, Sameer Singh, and Carlos Guestrin. "" Why should i trust you?" Explaining the predictions of any classifier." Proceedings of the 22nd ACM SIGKDD international conference on knowledge discovery and data mining. 2016.
> > > - [7] Chris Russell. Efficient search for diverse coherent explanations. In Proceedings of the Conference on Fairness, Accountability, and Transparency, pp. 20–28, 2019.
> > > - [8] Zhao, Xingyu, et al. "Baylime: Bayesian local interpretable model-agnostic explanations." arXiv preprint arXiv:2012.03058 (2020).
> > > - [9] Guo, Wenbo, et al. "Lemna: Explaining deep learning based security applications." Proceedings of the 2018 ACM SIGSAC Conference on Computer and Communications Security. 2018.

---

### Official Review · Reviewer_A84V · 2021-11-04

**Correctness:** 3
**Technical Novelty And Significance:** 3
**Empirical Novelty And Significance:** 3
**Recommendation:** 5
**Confidence:** 3

**Main Review:**

Overall I found the paper interesting and relevant. Some related literature is missing which I've highlighted below. The experimental results mostly correctly demonstrate the methodological contributions. In my opinion, the definitions and notation would need another pass of re-writing as in the current version it was quite difficult to follow with multiple read-throughs. I would be interested to read more justification of the assumed setting (in particular, on the distributional parameters, and how the method can be extended to nonlinear settings).

Sporadic (hopefully actionable) feedback and comments below
* [Sec 1, par 1] "Contrastive" explanations were first introduced by Tim Miller [A]
* [Sec 1, par 1] CFEs cannot be directly used as "directive actions" due to absence of causal understanding of actions and consequences (see Karimi et al., 2020b, [B] and others)
* [Sec 1, par 3] On "Counterfactual Plans", perhaps the authors would refrain from introducing yet another terminology to prevent confusion. Perhaps it better to simply call them collections of CFEs? This vagueness presents itself throughout the paper, e.g., in paragraph 3: "the plan should be valid: by committing to any action ...". If I've understood correctly (a page later, in the General Setup), plans are collections of actions, and each action is a simultaneous change over multiple features. In this case, one could consider each action as "plan" itself, no? Also, CFEs aren't actions because of lack of causality (see earlier point).
* [Sec 1, par 5] the two examples seem identical: change in demographic population and covariate distribution
* [Sec 1, par 6] remove "easily", and cite [B]
* [Sec 1, def 1.1] what is the difference between "feasibility" and "validity?"
* [Sec 1, notation] perhaps consider the more common notation of bold letters for vectors, e.g., $\mathbf{x}_j \in \mathbb{R}^d$ for a specific CFE, becuase $x_j$ can be interpreted as one element of a CFE.
* [Sec 2] why is the Gelbrich distance considered, as opposed to other distances? Also, can the authors please add intuition as to what it means for the distance in distribution (mean, covariance) of model parameters to be bounded up to a distance of $\rho$? Besides technical convenience, perhaps some intuitive or methodological justification can be provided?
* [Sec 2; nit] if $\tilde{\theta}$ has a nominal distribution $\hat{\mathbb{P}}$ with first and second moments $\hat{\mu}, \hat{\Sigma}$, wouldn't it be more consistent (and easier to read) if the model parameters were instead $\hat{\theta}$?
* [Sec 2] $\mathcal{P}$ is undefined and lacks explanations. It is also not used later, at least not directly ($\mathbb{B}$ is used). Generally, the notation is difficult to follow, as was the definition of CFE vs CF plans, etc.
* [Sec 2] under what conditions is $\Theta$ (not) empty?
* [Sec 2] abuse of notation in $\mathbb{Q}$ should be addressed, e.g., $\mathbb{Q}, \mathbb{Q} \sim (\cdot, \cdot), \mathbb{Q}(\cdot \in \cdot)$
* [Sec 2] evaluating the lower/upper bound of the probabilities of validity is reduced to solving a (rather complex) semi-definite program; is this easily solvable and are tradeoffs to be made here?
* [Sec 2] perhaps the authors want to clarify how $L^\star, U^\star$ act as a diagnostic tool, even with simple examples to aid readability.
* [Sec 3] abuse of notation in $\Theta({x_j})$
* [Sec 3] $\lambda_j^\star$ undefined
* [Sec 5] besides $\Sigma_g = (1 + \beta)I$, what other types of shifts in covariance could/should one consider? This seems like a rather restricted setting.
* [Sec 5, closing remarks] are there any comments on how the presented framework would extent to support nonlinear decision boundaries or other types of shifts in parameter distribution?


[A] "Contrastive Explanation: A Structural-Model Approach", Miller, 2018
[B] "The philosophical basis of algorithmic recourse", Venkatasubramanian & Alfano, 2020
[C] "On Counterfactual Explanations under Predictive Multiplicity", Pawelczyk et al., 2020

**Summary Of The Paper:**

This paper studies the effect of uncertainly in the parameters of an ML model used for consequential decision making on the validity, proximity, and diversity of sets of counterfactual explanations (CFE), namely counterfactual plans. In particular, the paper assumes the model parameters are sampled from a distribution would known mean and covariance, and studies the aforementioned metrics of counterfactual plans when the parameter distribution changes in a bounded manner (according to the Gelbrich distance). Beyond a diagnostic tool (providing bounds on how much the parametere distribution can change), the paper presents a method for counterfactual plan correction after having generated a plan (ex-post), and an approach to generate robust counterfactual plans before the fact (ex-ante). Experiments are presented for the proposed methods.

**Summary Of The Review:**

Overall I found the paper interesting and relevant. Some related literature is missing which I've highlighted below. The experimental results mostly correctly demonstrate the methodological contributions. In my opinion, the definitions and notation would need another pass of re-writing as in the current version it was quite difficult to follow with multiple read-throughs. I would be interested to read more justification of the assumed setting (in particular, on the distributional parameters, and how the method can be extended to nonlinear settings).

---

> ### Author Response · Authors · 2021-11-22
> **Response to Reviewer A84V [Part 1/3]**
>
> Thank you for the insightful comments. We appreciate your interest in our paper. Below we address the reviewer’s questions:
>
> **(1) Minor**: Thank you for many constructive suggestions. We have included the recommended changes in the main paper.
> * Misleading definitions: We have revised the entire paper and used only two terms "recourse" and "counterfactual" (or counterfactual explanation) interchangeably throughout the paper. By this definition, a recourse/counterfactual refers to a single vector $x_j$ and a counterfactual plan is a collection of recourses/counterfactuals $\{x_j\}$. Similar to “feasibility” and “validity”, we now only use validity throughout the entire paper for consistency.
> * Notations: We agree that some notations and definitions are hard to follow. Thus we have revised the manuscript and made changes according to your comments. Now, only $\Theta(\\{x_j\\})$ is used to show the dependence on the counterfactual plan $\{x_j\}$. We have also added explanations for undefined notations.
> * Citations: Thank you for the suggestion, we have included the recommended works in the revised draft.
> * [Sec 1, notation] bold letters for vectors: Most notations in this paper are vectors, and we have revised thoroughly to indicate the dimensions of each introduced notation. Besides, all $x_j$ notations refer to a vector, we avoid referring to a single feature (a specific element in the vector $x_j$) to prevent confusion.
> * [Sec 2; nit]: using $\hat{\theta}$ instead of $\hat{\mu}$: Another notation $\hat{\mu}$ is used to preserve the correspondence and avoid confusion between symbols since both $\mu$ and $\hat{\mu}$ are used (e.g. Theorem 2.2).
>
>
> **(2) Gelbrich distance and intuition behind the uncertainty set**: The primary goal of this work is to generate a counterfactual plan that is robust to changes in the model’s parameters. So that we define the uncertainty of those parameters by a distribution parameterized by $(\hat{\mu}, \Sigma)$. However, it is hard to estimate $(\hat{\mu}, \hat{\Sigma})$ correctly in practical use. By defining the uncertainty set $\mathcal{U}$ of mean and covariance bounded up to a Gelbrich distance of $\rho$, our formulation could hedge against the misspecification of these parameters.
>
> Additionally, thanks to the choice of the Gelbrich distance $\mathbb{G}$, both optimization problems in Theorems 2.2 and 2.3 are *linear* semidefinite programs, and they can be solved efficiently by standard, off-the-shelf solvers such as MOSEK to high dimensions. Other choices of distance (divergence) are also available: for example, one may opt for the Kullback-Leibler (KL) type divergence between Gaussian distribution to prescribe $\mathcal{U}$ as in Taskesen et al. [7]. Unfortunately, the KL divergence will entail a log-determinant term, and the resulting optimization problems are no longer linear programs and are no longer solvable using MOSEK.
>
> **(3) Trade-off in solving semidefinite programming**: While the optimization problems that define $L^\star$ and $U^\star$ may look complex, they are actually linear semidefinite programs, and they can be readily solved by conic solvers such as SDPT3/SeDuMi/MOSEK. State-of-the-art conic solvers use interior point method and have polynomial complexity. Specialized algorithms can also be exploited to improve the solution time (see Jiang et al. [8], Porkolab et al. [9], Zheng et al. [10]) Thus, evaluating $L^\star$ and $U^\star$ can be efficiently done in high dimensions. For example, for a feature space of dimension 60 and with a counterfactual plan of size 10, solving for $L^\star$ can be done within minutes on a 4-year-old laptop using MOSEK.
>
> **(4) Using $L^\star$ and $U^\star$ as a diagnostic tool**: In practice, $L^\star$ and $U^\star$ provide the lower (pessimistic) and upper (optimistic) bound on the probability of a valid plan, and thus these two values provide the decision-maker with a predicted range for the validity of any constructed counterfactual plan. We have included a short discussion after Theorem 2.3 to specify this relationship
> $  L^\star \leq \mathbb{Q}(\{x_j\} \text{ is a valid plan}) \leq U^\star \qquad \forall \mathbb{Q} \in \mathbb{B}. $
>
> A by-product of solving for $L^*$ is the values of the dual variables $\lambda^\star$, which can be exploited to improve/increase the lower bound on the validity probability as we have described in Section 3 of our paper.
>
> **(5) When is $\Theta$ empty?**: In most practical applications, the linear classifiers usually have the bias term so that $C_\theta(x)  = 1$ if $\theta^{T}  [1, x] \geq 0$ and 0 otherwise. In those cases, the set $\Theta$ is nonempty. Indeed, for an arbitrary plan $\{x_j\}$, there always exist a $\theta'$ such that $\theta_0' \geq  max_{j \in [J]} -{\theta_{1:}'}^{T} x_j$ where $\theta_0'$ is the intercept and $\theta_{1:}'$ is the coefficients of the classifier $\mathcal{C}_{\theta'}$.

---

> > ### Author Response · Authors · 2021-11-22
> > **Response to Reviewer A84V [Part 2/3]**
> >
> > **(6) Types of covariance shift Section 5**: The main purpose of this experiment is to investigate the impact of the degree of the distribution shift on the validity of the plans. So we parameterize $\Sigma_g = (1+\beta)I$ for the sake of simplicity. For further investigation, we have included an additional experiment in Appendix B.2 with a different covariance shift so that the positive and negative correlation between parameters are also introduced: $\Sigma_g = (1+\beta)A$, where
> > $
> >     A = \begin{pmatrix}
> >     1 & -1 & 0\\\\
> >     -1 & 1 & 1 \\\\
> >     0 & 1 & 1
> >     \end{pmatrix}
> > $
> >
> > **(7) Extension to nonlinear classifiers**: Similar to some previous works (Ustun et al [3], Rawal et al. [1], Upadhyay et al. [2]), our work can extend to nonlinear classifiers by using a local linear model that approximates the decision boundary around a given input instance (e.g. LIME [6]). To evaluate this approach, we have conducted an experiment with a nonlinear classifier (three-layer MLP) in three real-world datasets. We have included this experiment in Appendix C.2 of the paper. We report the table below:
> >
> > |Dataset        |Method                                       |Proximity               |Diversity                |$L^*$                |Empirical Validity                 |Current Validty     |
> > |----------|-----------------------------------------|---------------------|---------------------|---------------------|---------------------|---------|
> > |Correction|DiCE                                     |0.515 $\pm$ 0.204    |0.043 $\pm$ 0.037    |0.005 $\pm$ 0.041    |0.414 $\pm$ 0.238    |**0.990**|
> > |          |MahalanobisCrr                           |0.595 $\pm$ 0.210    |0.035 $\pm$ 0.035    |0.021 $\pm$ 0.058    |0.409 $\pm$ 0.313    |0.670    |
> > |          |COPA ($\lambda_1 = 0.1; \lambda_2 = 1.0$)|**0.219** $\pm$ 0.183|0.001 $\pm$ 0.011    |**0.065** $\pm$ 0.088|**0.556** $\pm$ 0.331|0.560    |
> > |          |COPA ($\lambda_1 = 0.1; \lambda_2 = 2.0$)|0.432 $\pm$ 0.403    |0.100 $\pm$ 0.116    |0.049 $\pm$ 0.093    |0.301 $\pm$ 0.341    |0.270    |
> > |          |COPA ($\lambda_1 = 0.2; \lambda_2 = 2.0$)|0.673 $\pm$ 0.314    |**0.162** $\pm$ 0.084|0.038 $\pm$ 0.097    |0.125 $\pm$ 0.186    |0.040    |
> > |Temporal  |DiCE                                     |1.573 $\pm$ 0.451    |0.107 $\pm$ 0.071    |0.637 $\pm$ 0.350    |0.852 $\pm$ 0.270    |**1.000**|
> > |          |MahalanobisCrr                           |1.567 $\pm$ 0.449    |0.099 $\pm$ 0.070    |0.868 $\pm$ 0.118    |0.987 $\pm$ 0.076    |**1.000**|
> > |          |COPA ($\lambda_1 = 0.1; \lambda_2 = 1.0$)|**1.388** $\pm$ 0.540|0.002 $\pm$ 0.008    |0.981 $\pm$ 0.014    |**1.000** $\pm$ 0.000|**1.000**|
> > |          |COPA ($\lambda_1 = 0.1; \lambda_2 = 2.0$)|1.534 $\pm$ 0.408    |**0.247** $\pm$ 0.043|0.976 $\pm$ 0.012    |**1.000** $\pm$ 0.000|**1.000**|
> > |          |COPA ($\lambda_1 = 0.2; \lambda_2 = 2.0$)|1.447 $\pm$ 0.340    |0.118 $\pm$ 0.072    |**0.990** $\pm$ 0.004|**1.000** $\pm$ 0.000|**1.000**|
> > |Geospatial|DiCE                                     |1.576 $\pm$ 0.349    |0.175 $\pm$ 0.070    |0.022 $\pm$ 0.046    |0.328 $\pm$ 0.303    |**1.000**|
> > |          |MahalanobisCrr                           |1.594 $\pm$ 0.349    |0.169 $\pm$ 0.071    |0.113 $\pm$ 0.084    |**0.689** $\pm$ 0.280|**1.000**|
> > |          |COPA ($\lambda_1 = 0.1; \lambda_2 = 1.0$)|**1.342** $\pm$ 0.367|0.000 $\pm$ 0.000    |0.011 $\pm$ 0.007    |0.384 $\pm$ 0.310    |0.710    |
> > |          |COPA ($\lambda_1 = 0.1; \lambda_2 = 2.0$)|1.552 $\pm$ 0.292    |0.243 $\pm$ 0.039    |0.010 $\pm$ 0.024    |0.168 $\pm$ 0.210    |0.750    |
> > |          |COPA ($\lambda_1 = 0.2; \lambda_2 = 2.0$)|1.637 $\pm$ 0.284    |**0.287** $\pm$ 0.017|**0.164** $\pm$ 0.066|0.679 $\pm$ 0.274    |**1.000**|
> >
> > The result is promising since the proposed methods can increase the empirical validity significantly. However, the infidelity of LIME could lead to invalid counterfactual explanations, represented by a lower current validity value. The low current validity is also observed in the literature, see Upadhyay et al. (2021). For further investigation, one can use another local surrogate model that provides a better approximation of the decision boundary (e.g. BayLIME [11]). Another direction that should be considered is to use a mixture linear regression model to approximate the decision boundary as in Guo et al [12]. However, further analysis has to be made for the mixture linear model.

---

> > > ### Author Response · Authors · 2021-11-22
> > > **Response to Reviewer A84V [Part 3/3]**
> > >
> > > We hope that we have addressed all your concerns adequately. We have updated the manuscript according to your comments. Please let us know if we can provide any further details and/or clarifications.
> > >
> > >
> > > *References*:
> > > - [1] Rawal, Kaivalya, Ece Kamar, and Himabindu Lakkaraju. "Can I Still Trust You?: Understanding the Impact of Distribution Shifts on Algorithmic Recourses." arXiv:2012.11788 (2020).
> > > - [2] Upadhyay, Sohini, Shalmali Joshi, and Himabindu Lakkaraju. "Towards Robust and Reliable Algorithmic Recourse." arXiv preprint arXiv:2102.13620 (2021).
> > > - [3] Ustun, Berk, Alexander Spangher, and Yang Liu. "Actionable recourse in linear classification." Proceedings of the Conference on Fairness, Accountability, and Transparency. 2019.
> > > - [4] Karimi, Amir-Hossein, Bernhard Schölkopf, and Isabel Valera. "Algorithmic recourse: from counterfactual explanations to interventions." Proceedings of the 2021 ACM Conference on Fairness, Accountability, and Transparency. 2021.
> > > - [5] Karimi, Amir-Hossein, et al. "Algorithmic recourse under imperfect causal knowledge: a probabilistic approach." arXiv preprint arXiv:2006.06831 (2020).
> > > - [6] Ribeiro, Marco Tulio, Sameer Singh, and Carlos Guestrin. "" Why should i trust you?" Explaining the predictions of any classifier." Proceedings of the 22nd ACM SIGKDD international conference on knowledge discovery and data mining. 2016.
> > > - [7] Taskesen, Bahar, et al. "Sequential Domain Adaptation by Synthesizing Distributionally Robust Experts." arXiv preprint arXiv:2106.00322 (2021).
> > > - [8] Jiang, Haotian, et al. "A faster interior point method for semidefinite programming." 2020 IEEE 61st Annual Symposium on Foundations of Computer Science (FOCS). IEEE, 2020.
> > > - [9] Porkolab, Lorant, and Leonid Khachiyan. "On the complexity of semidefinite programs." Journal of Global Optimization 10.4 (1997): 351-365.
> > > - [10] Zheng, Yongbin, et al. "An efficient approach to solve the large-scale semidefinite programming problems." Mathematical Problems in Engineering 2012 (2012).
> > > - [11] Zhao, Xingyu, et al. "Baylime: Bayesian local interpretable model-agnostic explanations." arXiv preprint arXiv:2012.03058 (2020).
> > > - [12] Guo, Wenbo, et al. "Lemna: Explaining deep learning based security applications." Proceedings of the 2018 ACM SIGSAC Conference on Computer and Communications Security. 2018.

---

> > > > ### Author Response · Authors · 2021-11-30
> > > > **Response to Reviewer A84V - For any further questions**
> > > >
> > > > Dear reviewer,
> > > >
> > > > We thank the reviewer again for your insightful reviews and thoughtful suggestions. We have done our best to address the questions raised in your review. The discussion period is coming to a close within a day and we remain open to discussing any remaining concerns you may have until the very end. If there are any remaining questions or concerns, please do not hesitate to ask.
> > > >
> > > > Thank you for taking the time to evaluate our work and our responses.
> > > >
> > > > Authors

---

### Decision · Program_Chairs · 2022-01-20

**Decision:**

Accept (Poster)

**Comment:**

The paper provides a neat idea about explaining (linear) predictors based on designing ways of perturbing parameters. It is focused on linear models (which can still lead to non-linear classifiers), but it is a relevant case, particularly for explainability.